# Efficacy of epetraborole against *Mycobacterium abscessus* is increased with norvaline

**Jaryd R. Sullivan**[1,2,3], **Andréanne Lupien**[2,3], **Elias Kalthoff**[4,5], **Claire Hamela**[6], **Lorne Taylor**[7], **Kim A. Munro**[4,5], **T. Martin Schmeing**[4,5], **Laurent Kremer**[6,8], **Marcel A. Behr**[1,2,3,9]*

**1** Department of Microbiology & Immunology, McGill University, Montréal, Canada, **2** Infectious Diseases and Immunity in Global Health Program, Research Institute of the McGill University Health Centre, Montréal, Canada, **3** McGill International TB Centre, Montréal, Canada, **4** Department of Biochemistry, McGill University, Montréal, Canada, **5** Centre de Recherche en Biologie Structural, McGill University, Montréal, Canada, **6** Centre National de la Recherche Scientifique UMR 9004, Institut de Recherche en Infectiologie de Montpellier (IRIM), Université de Montpellier, Montpellier, France, **7** Clinical Proteomics Platform, Research Institute of the McGill University Health Centre, Montréal, Canada, **8** INSERM, IRIM, Montpellier, France, **9** Department of Medicine, McGill University Health Centre, Montréal, Canada

\* marcel.behr@mcgill.ca

**Data Availability Statement:** All relevant data are within the paper and its Supporting Information files. Structures have been deposited on PDB under accession codes 7N11 and 7N12. Sequence

## Abstract

*Mycobacterium abscessus* is the most common rapidly growing non-tuberculous mycobacteria to cause pulmonary disease in patients with impaired lung function such as cystic fibrosis. *M. abscessus* displays high intrinsic resistance to common antibiotics and inducible resistance to macrolides like clarithromycin. As such, *M. abscessus* is clinically resistant to the entire regimen of front-line *M. tuberculosis* drugs, and treatment with antibiotics that do inhibit *M. abscessus* in the lab results in cure rates of 50% or less. Here, we identified epetraborole (EPT) from the MMV pandemic response box as an inhibitor against the essential protein leucyl-tRNA synthetase (LeuRS) in *M. abscessus*. EPT protected zebrafish from lethal *M. abscessus* infection and did not induce self-resistance nor against clarithromycin. Contrary to most antimycobacterials, the whole-cell activity of EPT was greater against *M. abscessus* than *M. tuberculosis*, but crystallographic and equilibrium binding data showed that EPT binds LeuRS$_{Mabs}$ and LeuRS$_{Mtb}$ with similar residues and dissociation constants. Since EPT-resistant *M. abscessus* mutants lost LeuRS editing activity, these mutants became susceptible to misaminoacylation with leucine mimics like the non-proteinogenic amino acid norvaline. Proteomic analysis revealed that when *M. abscessus* LeuRS mutants were fed norvaline, leucine residues in proteins were replaced by norvaline, inducing the unfolded protein response with temporal changes in expression of GroEL chaperonins and Clp proteases. This supports our *in vitro* data that supplementation of media with norvaline reduced the emergence of EPT mutants in both *M. abscessus* and *M. tuberculosis*. Furthermore, the combination of EPT and norvaline had improved *in vivo* efficacy compared to EPT in a murine model of *M. abscessus* infection. Our results emphasize the effectiveness of EPT against the clinically relevant cystic fibrosis pathogen *M. abscessus*, and these findings also suggest norvaline adjunct therapy with EPT could be beneficial for *M. abscessus* and other mycobacterial infections like tuberculosis.

data has been deposited in the NCBI Sequence Read Archive under the NCBI BioProject ID PRJNA756101. (https://www.ncbi.nlm.nih.gov/sra/X).

**Funding:** This work was funded by Cystic Fibrosis Canada (to J.R.S. and M.A.B.), the Research Institute of the McGill University Health Centre (to J.R.S.), the Harrison Watson Fellowship (to J.R.S.), the Structure-guided Drug Discovery Coalition (to M.A.B.), CIHR operating grant (to M.A.B.) and was supported by Vaincre la Mucoviscidose (RF20200502678) (to L.K) and the Association Gregory Lemarchal (to L.K). The funders had no role in study design, data collection and analysis, decision to publish, or preparation of the manuscript.

**Competing interests:** The authors have declared that no competing interests exist.

## Author summary

Current antimycobacterial drugs are inadequate to handle the increasing number of non-tuberculous mycobacteria infections that eclipse tuberculosis infections in many developed countries. Of particular importance for cystic fibrosis patients, *Mycobacterium abscessus* is notoriously difficult to treat where patients spend extended time on antibiotics with cure rates comparable to extreme drug resistant *M. tuberculosis*. Here, we identified epetraborole (EPT) with *in vitro* and *in vivo* activities against *M. abscessus*. We showed that EPT targets the editing domain of the leucyl-tRNA synthetase (LeuRS) and that escape mutants lost LeuRS editing activity, making these mutants susceptible to misaminoacylation with leucine mimics. Most importantly, combination therapy of EPT and norvaline limited the rate of EPT resistance in both *M. abscessus* and *M. tuberculosis*, and this was the first study to demonstrate improved *in vivo* efficacy of EPT and norvaline compared to EPT in a murine model of *M. abscessus* pulmonary infection. The demonstration of norvaline adjunct therapy with EPT for *M. abscessus* infections is promising for cystic fibrosis patients and could translate to other mycobacterial infections, such as tuberculosis.

## Introduction

*Mycobacterium abscessus* is a nontuberculous mycobacterium that commonly causes chronic lung disease, especially among patients with cystic fibrosis [1]. Treatment of pulmonary exacerbation relies on a regimen of intravenous amikacin (AMK), tigecycline, and imipenem plus an oral macrolide if the isolate is susceptible to macrolides [2,3]. Despite guideline-based treatment, combination therapy typically results in cure rates of 50% or less due to the bacterium being intrinsically resistant to many antibiotic classes [4,5]. Unfortunately, the *M. abscessus* drug-development pipeline is limited to a handful of repurposed drugs (clofazimine, rifabutin, and bedaquiline) [6]. Lately, antibiotics that have shown activity in pre-clinical studies include the oxazolidinones (LCB01-0371 [7] and tedizolid [8]) that target the 50S ribosomal subunit and PIPD1/indol-2-carboxamides that target the mycolic acid transporter MmpL3 [9,10].

Recently, aminoacyl-tRNA synthetases (aaRSs) became targets of interest for drug discovery when benzoxaboroles were identified as novel boron-based pharmacophores [11,12]. aaRSs are enzymes that aminoacylate tRNAs with their cognate amino acids. All aaRSs have an aminoacylation domain while some are bifunctional and contain an editing domain. The aminoacylation domain forms an aminoacyl adenylate through condensation of the amino acid with ATP, and then transfers the aminoacyl moiety to the 3' terminal adenosine of the tRNA acceptor stem. Some aaRSs, however, have evolved an editing domain to ensure the correct amino acid is ligated to its tRNA [13,14]. This domain is critical for aaRSs that must distinguish their cognate amino acid from structurally similar amino acids like branched-chain amino acids. Leucyl-tRNA synthetases (LeuRSs) are examples of aaRSs that rely on their editing domains to limit misaminoacylation of tRNA with near-cognate amino acids and maintain the fidelity of the genetic code.

Herein, we identified epetraborole (EPT) from the Medicines for Malaria Venture (MMV) Open pandemic response box as a LeuRS inhibitor in *M. abscessus* with nanomolar activity *in vitro* and activity in zebrafish embryos and NOD.SCID mice. Interestingly, EPT was more active against *M. abscessus* than *M. tuberculosis*, and this observation was not supported by

structural differences in the crystal structures of the LeuRS editing domain of *M. abscessus* and *M. tuberculosis* bound to EPT. Furthermore, we highlighted the utility of norvaline to target LeuRS editing deficient escape mutants and suppress resistance *in vitro* by misincorporation of norvaline in place of leucine residues resulting in disrupted protein folding. Importantly, we showed that EPT and norvaline combination has improved efficacy over EPT monotherapy in a murine model of *M. abscessus* infection. These data support the potential of the benzoxaborole scaffold in the antimycobacterial drug pipeline and suggest that its activity can be potentiated by supplementation with norvaline or a norvaline derivative.

## Results

### Discovery of an antimycobacterial inhibitor against *M. abscessus*

To identify compounds with antimycobacterial activity, we first engineered a luminescent *M. abscessus* ATCC 19977 strain which constitutively expressed the *luxCDABE* operon [15] (*M. abscessus lux*). We then screened the 176 compound open-library provided by GlaxoSmithKline (S1A–S1C Fig) and the 400 compound Pathogen and Pandemic Response Boxes from MMV (S1D–S1F Fig) using a threshold $\geq$ 90% reduction in luminescence compared to non-treated bacteria at 10 μM. Primary hits (20 compounds) were tested in a secondary screen using the resazurin microtiter assay (REMA) at 10 μM on the *M. abscessus* ATCC 19977 strain. Secondary hits (9 compounds) were then tested in a dose-response assay on *M. abscessus* ATCC 19977 from a fresh solid compound to determine $MIC_{90}$. The HepG2 (human hepatocytes) cell line was used to assess the toxicity of hits. Among three compounds (0.3% positive hits of 976 compounds) from the pandemic response box that satisfied primary and secondary criteria, we identified EPT (Fig 1A) as having the most potent antimycobacterial activity and highest therapeutic index (Table 1, $MIC_{90}$ 0.23 μM (0.063 μg/mL), TI ($TD_{50}/MIC_{90}$) EPT > 330). It was previously reported in *M. tuberculosis* that potent *in vitro* growth inhibitors could display carbon-source-dependent effects, which leads to a loss of activity when advanced into *in vivo* models [16]. We thus measured the $MIC_{90}$ of EPT on different carbon sources (glycerol vs acetate), in the presence or absence of the detergent Tween-80, and in different nutrient bases (Middlebrook 7H9 vs cation-adjusted Mueller-Hinton). EPT was active in all assayed growth conditions but lost potency in cation-adjusted Mueller Hinton (CaMH) media (S1 Table). Lower activity of EPT in CaMH media is unsurprising as it was previously shown that rifamycins [17] as well as other antimicrobials [18] lose activity in CaMH media (0.5mg/L $Ca^{2+}$ and 0.05mg/L $Mg^{2+}$ in 7H9; 20-25mg/L $Ca^{2+}$ and 10–12.5mg/L $Mg^{2+}$ in CaMH). It is thought that some antimicrobials may chelate divalent metal ions, thus limiting their uptake in cells.

Next, we tested the *in vitro* activity of EPT against a panel of *M. abscessus* clinical isolates belonging to the three subspecies of the *M. abscessus* complex (*M. abscessus*, *M. massiliense*, and *M. bolletii*) with smooth and rough colony morphologies and with different drug susceptibility profiles to various antibiotics. There was no loss in activity against clinical isolates (range of 0.014–0.046 μg/mL) nor different morphologies (S2 Table). To determine the spectrum of activity of EPT, we curated a panel of various mycobacteria and representative gram-positive and gram-negative bacteria. Interestingly, EPT appears to be more selective for *M. abscessus* with lower activity against *M. tuberculosis* H37Rv (S3 Table). *In vitro* growth kill kinetics indicated that EPT is bacteriostatic against *M. abscessus* as previously demonstrated against *M. tuberculosis* [19], while 10% of CFUs were lost at 24 hours with AMK and rifampicin (RIF) at 20X $MIC_{90}$. Interestingly, these two antimycobacterial agents with bactericidal action against other mycobacteria lose this ability when targeting *M. abscessus* (Fig 1B) [20].

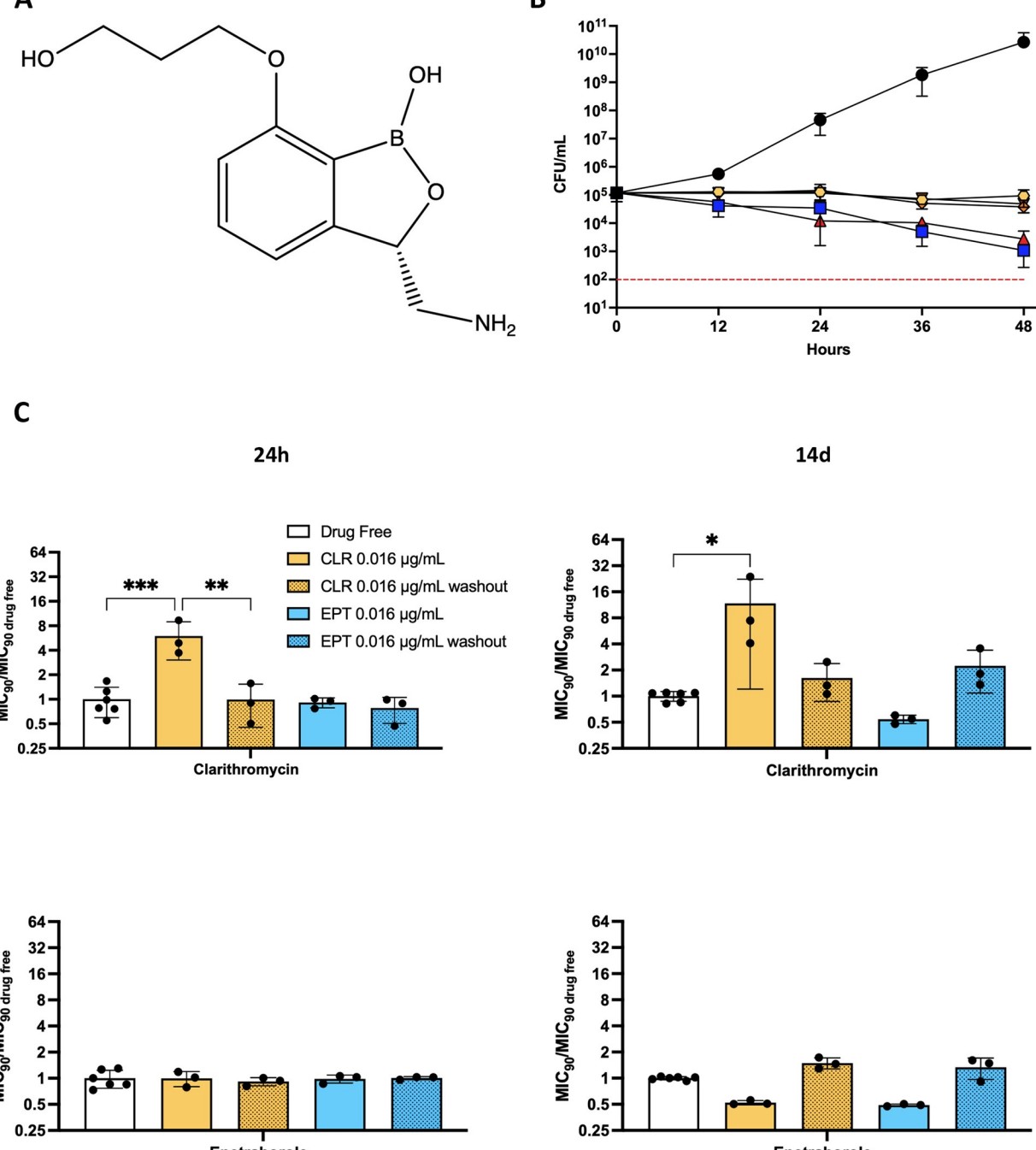

**Fig 1. EPT *in vitro* activity against *M. abscessus*. A** Structure of EPT. **B** Epetraborole kill kinetics of *M. abscessus in vitro*. Bacteria were incubated in 7H9 complete with drug-free (black circles), AMK 20X $MIC_{90}$ (blue squares, 80 µg/mL), RIF 20X $MIC_{90}$ (red triangles, 60 µg/mL), EPT 20X $MIC_{90}$ (yellow hexagons, 1.6 µg/mL), EPT 40X $MIC_{90}$ (orange diamonds, 3.2 µg/mL), or EPT 80X $MIC_{90}$ (brown inverted triangles, 6.4 µg/mL). The dashed red line indicates 99.9% or 3 $\log_{10}$ reduction in CFU/mL threshold of a bactericidal drug. Data shown is mean ± SD of three independent experiments. **C** $MIC_{90}$ from induced cultures relative to drug free conditions. Bacteria were grown to $OD_{600}$ 0.05 and induced with no drug (white), 0.016 µg/mL CLR (yellow), 0.016 µg/mL EPT (blue). After 24 hours (left) or 14 days (right) of induction, cultures were passaged for 6 days in drug free media (dotted bars). Data shown is mean ± SD from three independent experiments. Error bars for drug free data derived from $MIC_{90}$ / $MIC_{90}$ average from three or six replicates. *, p = 0.02; **, p = 0.0017; ****, p = 0.0005 by one-way ANOVA with Tukey's multiple comparisons test.

**Table 1. MMV Open pandemic response box hits and reference compounds.**

| Parameter | Compound | | | | |
|---|---|---|---|---|---|
| | EPT | ERV | IQN | BDQ | AMK |
| $MIC_{90}$ (μg/mL) | 0.063 | 1.9 | 3.3 | 0.67 | 4.0 |
| $TD_{50}$ (μg/mL) | >27 | >63 | >39 | 9.4[a] | 18 |
| TI ($TD_{50}/MIC_{90}$) | >430 | 33 | 12 | 14 | 4.5 |
| MW (g/mol) | 237.06 | 631.52 | 393.40 | 555.51 | 585.60 |

EPT, Epetraborole; ERV, Eravacycline; IQN, Isoquinoline urea; BDQ, Bedaquiline; AMK, Amikacin; $MIC_{90}$, concentration of drug that inhibits 90% of bacterial growth; $TD_{50}$, concentration of drug with 50% toxicity against HepG2 cell line. TI, therapeutic index.

[a]Data from Lupien, A et al. *Antimicrob Agents Chemother.* **2018**

## Clinical considerations for EPT

Macrolides represent the cornerstone of *M. abscessus* therapy. However, macrolide susceptibility *in vitro* does not correlate with clinical outcome success due to point mutations at positions 2058 or 2059 in the 23S rRNA *rrl* gene (*E. coli* numbering), and inducible macrolide resistance conferred from the recently identified ribosomal methyltransferase *erm*(41) [21,22]. Nash et al (2009) discovered that *M. abscessus* exhibited an inducible resistant phenotype to macrolides, like clarithromycin (CLR), during a 14-day incubation. To this extent, we asked if EPT induced self-resistance or cross-resistance to macrolides. To answer this question, we performed the inducible macrolide resistance assay where *M. abscessus* was cultured with a subinhibitory concentration of CLR (0.016 μg/mL) or EPT (0.016 μg/mL) for 14 days. At 24 hours and 14 days, aliquots of culture were collected, and the $MIC_{90}$ of CLR and EPT was determined using REMA. Relative to drug-free conditions, *M. abscessus* cultured with CLR displayed an increased $MIC_{90}$ to CLR while the EPT $MIC_{90}$ for bacteria exposed to EPT remained unchanged up to 14 days (Fig 1C solid bars). To ensure the increased $MIC_{90}$ resulted from inducible resistance rather than the selection of spontaneous CLR and EPT resistant mutants, the cultures were passaged in antibiotic-free media for 6 additional days and the $MIC_{90}$ measurements were repeated. In the case of CLR resistance, the $MIC_{90}$ returned to baseline after the antibiotics were removed (Fig 1C dotted bars). In addition, EPT stimulation did not result in cross-resistance to CLR, and CLR induction did not result in cross-resistance to EPT. The latter is an important finding with clinical significance because CLR is known to impart resistance to aminoglycoside antibiotics [23].

Because *M. abscessus* infections require 18–24 months of antibiotic courses, multidrug treatments are standard practice [2,3]. To ensure EPT would not hamper a multidrug regimen, we performed checkerboard assays with common antimycobacterial drugs. In the checkerboard assays, we used RIF/CLR and CLR/AMK as the synergy and antagonism controls, respectively [23,24]. We did not observe antagonism between EPT and various antimycobacterial agents that target a range of cellular processes (S2 Fig). Whether EPT should be included in a multidrug regimen remains to be determined.

## EPT protects zebrafish from *M. abscessus* infection

In order to investigate the *in vivo* activity of EPT, we used the embryonic zebrafish model of *M. abscessus* infection, which has been developed to test the *in vivo* efficacy of drugs against *M. abscessus* [25–27]. Initial experiments indicated that EPT concentrations up to 40 μg/mL (final concentration in fish water) did not interfere with larval development and was well tolerated in embryos when treatment was applied for up to 5 days with daily drug renewal (Fig 2A).

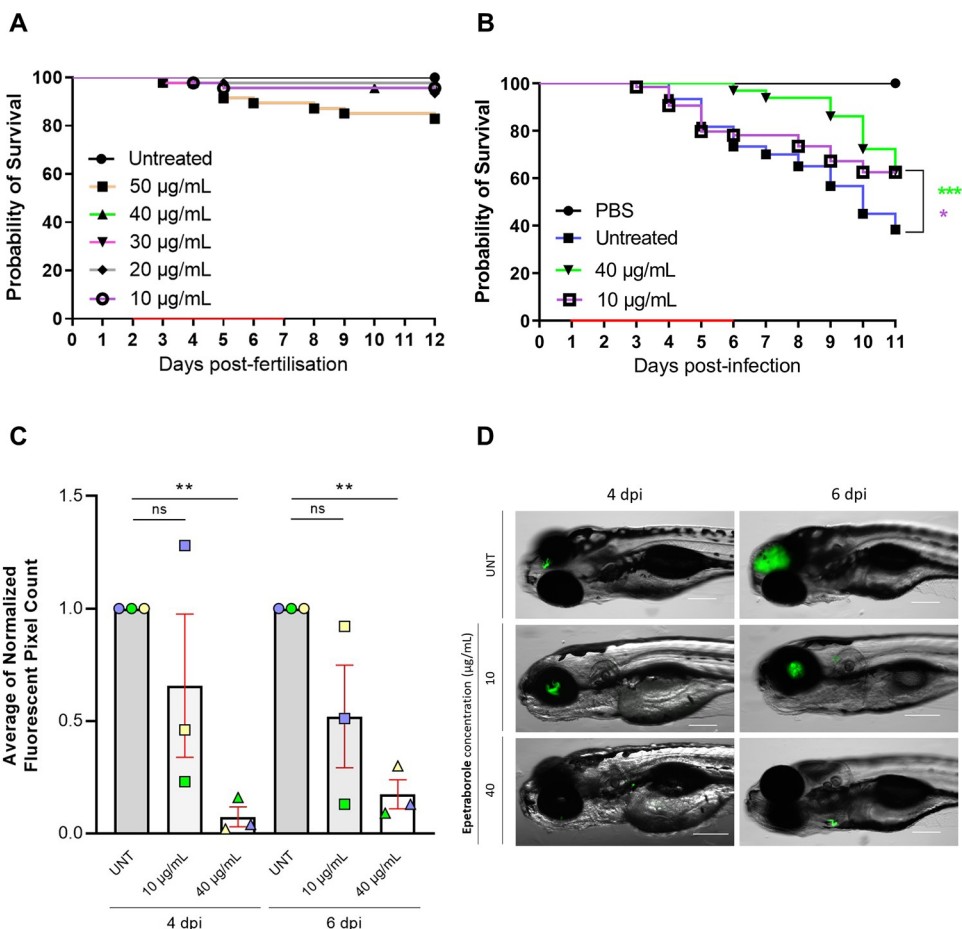

**Fig 2. EPT *in vivo* activity against *M. abscessus*. A** EPT toxicity against zebrafish embryos. Groups of uninfected embryos were immersed in water containing 10 to 50μg/mL EPT for 5 days. Red bar indicates duration of treatment. Data is from two independent experiments. **B** EPT *in vivo* activity in zebrafish embryos. Zebrafish embryos at 30 h postfertilization were infected with approximately 200 CFUs of *M. abscessus* expressing green fluorescent wasabi or PBS via caudal vein injection. At 1 dpi, embryos were randomly split into approximately 20 embryos per group and exposed to increasing concentrations of EPT (10 and 40 μg/mL) in fish water. Drugs were renewed at a daily basis for 5 days (red bar) after which embryos were washed twice in fresh embryo water and maintained in fish water. Each treatment group was compared against the untreated infected group with significant differences calculated using the log-rank (Mantel-Cox) statistical test for survival curves. Data shown are the merge of four independent experiments. *, p = 0.02; ***, p = 0.0007. **C** Fluorescent Pixel Counts (FPC) determination using the ImageJ software, reflecting the bacterial loads at 4 and 6 days post-infection (dpi). Each bar represents a pool of the average of normalized FPC from three independent experiments (n = 22 to 36 embryos). Error bars represent standard deviations. Statistical significance was determined using Welch's *t* test. **, p ≤ 0.0054. **D** Representative whole embryos from the untreated group (upper panels) and treated group (10 or 40 μg/mL EPT; lower panels) at 4 and 6 dpi. Green overlay represents *M. abscessus* expressing wasabi. Scale bar, 200 μm.

Green fluorescent wasabi-expressing *M. abscessus* (R variant) was microinjected in the caudal vein of embryos at 30 hours post-fertilization. EPT was directly added at 1 day post-infection to the water containing the infected embryos, with EPT-supplemented water changed on a daily basis for 5 days. Embryo survival was monitored and recorded daily for 11 days. While a slight increase in the survival rate was observed with 10 μg/mL EPT, this effect was significantly improved with 40 μg/mL EPT with a delay in larval mortality, as compared to the untreated group (Fig 2B). The protection provided by EPT is maintained throughout the 5-day treatment course and was correlated with decreased bacterial burdens beginning 4 dpi

demonstrated by fluorescent pixel counting (Fig 2C and 2D). These results clearly indicate that EPT protects zebrafish from *M. abscessus* infection.

## EPT targets LeuRS in *M. abscessus*

Benzoxaboroles were shown to inhibit LeuRS in fungi, gram-negative pathogens and most recently *M. tuberculosis* [11,19,28]. To determine the target of EPT in *M. abscessus*, $10^9$ CFUs were plated on solid media with 10X, 20X, or 40X $MIC_{90}$ of EPT to select EPT-resistant mutants. Control mutants were also selected on 40X MIC of AMK. Unlike the EPT resistance frequency of 4.8 x $10^{-8}$ in *Pseudomonas aeruginosa*, the resistance frequency was 10-fold lower in *M. abscessus* (S4 Table, 2 x $10^{-9}$) [28]. The low *in vitro* resistance frequency to EPT is highlighted when compared to the control mutants against AMK (1.3 x $10^{-8}$). In order to confirm that isolated resistant mutants were true mutants to EPT, we performed REMA using a panel of antimycobacterial agents on the four mutants (one selected on 10X, one on 20X, and two on 40X $MIC_{90}$ of EPT). We also screened one AMK mutant. As illustrated in S3 Fig, the four EPT mutants were resistant up to 2.7 µg/mL EPT, while maintaining susceptibility to AMK, bedaquiline (BDQ), and RIF. Likewise, the AMK control mutant was only resistant to AMK.

Since benzoxaboroles are known to target LeuRSs, we used a focused approach to identify the mutation(s) in *leuS* that could be responsible for the observed EPT resistance. *leuS* was PCR amplified from gDNA from the four EPT mutants, and single-nucleotide polymorphisms (SNPs) were identified using Sanger sequencing (S5 Table). In all four mutants, a G to C transversion at position 1306 was identified, which resulted in the conserved D436 residue critical for the catalysis of editing misaminoacylated tRNA^Leu substituted for H436 (Fig 3A and 3B) [29]. This was further supported with whole-genome sequencing of the *M. abscessus* ATCC 19977 reference strain, one mutant isolated at 20X MIC, and one mutant isolated at 40X MIC. When compared to the reference strain, five variants were identified in the 20X mutant including a T to G transition at position 1261 in *leuS* which resulted in a Y421D substitution. Furthermore, the G to C transversion at position 1306 which lead to the D436H substitution was confirmed in the 40X mutant (S6 Table). This contrasts with LeuRS variants identified in *P. aeruginosa* such as T323P, T327P, and V429M from Hernandez *et al.* or LeuRS variants such as T322I, D326N, A414V, and R435H identified in a phase 2 clinical trial for complicated urinary tract infections caused by *E. coli* from O'Dwyer *et al.* [28,30]. However, the phase 2 clinical trial did identify *E. coli* isolates with a LeuRS D436A variant. (*M. abscessus* numbering). To verify the functional significance of the D436H substitution, the *leuS* genes from *M. abscessus* ATCC 19977 wildtype (EPT sensitive) and *M. abscessus* containing the G1306C substitution (EPT-resistant) were cloned into the mycobacterial integrative vector pMV306 under the control of the constitutive *hsp60* promoter, yielding pMVhsp60_leuS^D and pMVhsp60_leuS^D436H. EPT-sensitive *M. abscessus* was electroporated with pMVhsp60_leuS^D, pMVhsp60_leuS^D436H, or empty vector control. Only EPT sensitive *M. abscessus* complemented with the mutant *leuS* (pMVhsp60_leuS^D436H) and not the wildtype *leuS* (pMVhsp60_leuS^D) had increased resistance to EPT (Fig 3C). The integrative complement system using pMVhsp60 may not have provided a complete phenocopy of the true mutant when compared to the episomal equivalent. Also, *M. abscessus* may not use the *hsp60* promoter from *M. bovis* with optimal efficiency. Lastly, the *hsp60* promoter may not operate with the same strength as the *leuS* promoter.

We showed that LeuRS^D436H confers high-level resistance to EPT and that complementing the resistant allele into a wildtype background imparts resistance. As a complementary means of verifying the target, we adapted a CRISPR-interference (CRISPRi) system for gene knockdown in *M. abscessus* [31]. CRISPRi facilitates gene knockdown using a guide RNA and catalytically inactive Cas9 endonuclease to sterically prevent transcription of a gene of interest

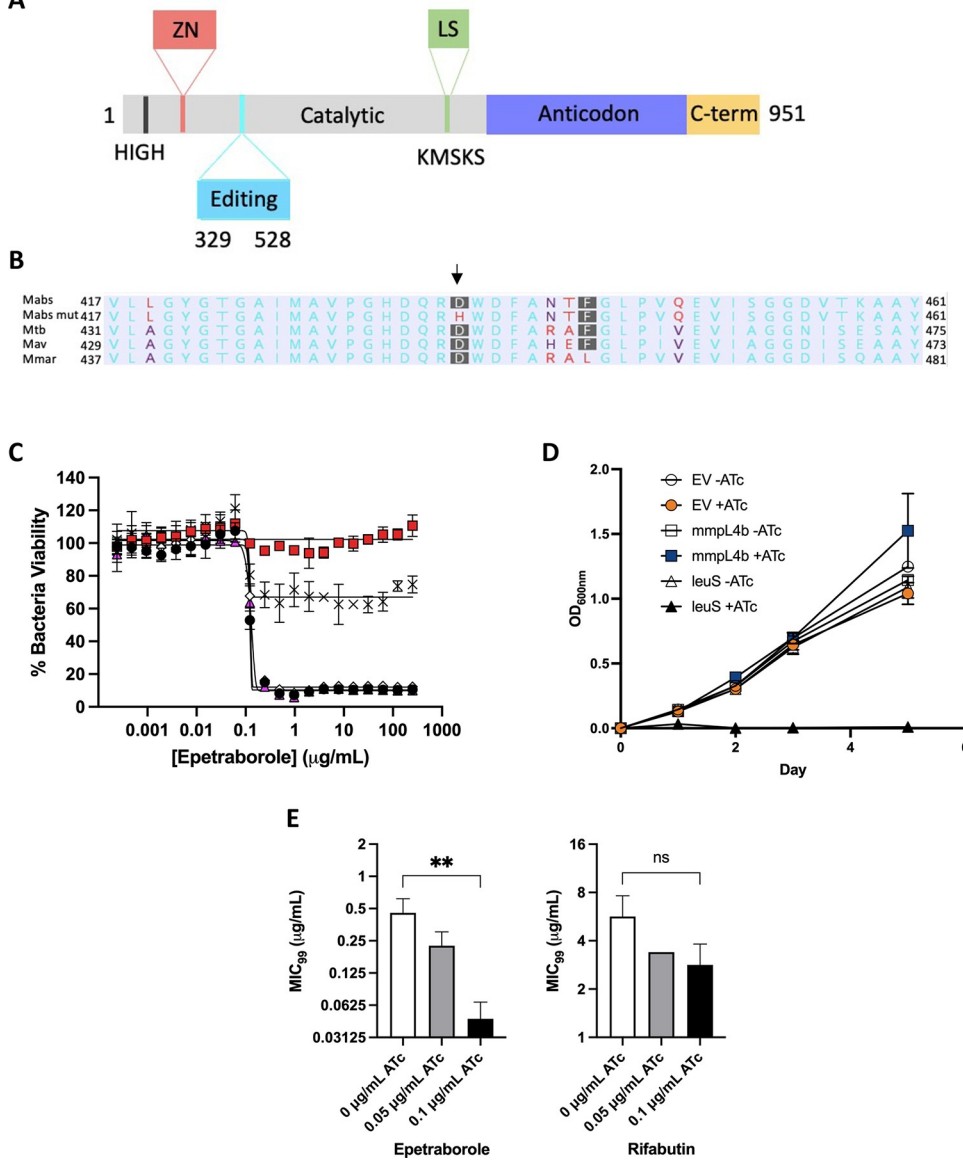

**Fig 3. EPT targets the editing domain of LeuRS in *M. abscessus*. A** Domains of *M. abscessus* LeuRS. Zinc domain, pink; editing domain, cyan; catalytic domain, grey; leucine-rich domain, green; anticodon-binding domain, purple; c-terminus, yellow. **B** Amino acid alignment of part of the editing domain of LeuRS *M. abscessus* ATCC 19977, EPT mutant, and reference mycobacteria. Non-consensus residues are in red. Genbank accession numbers: P9WFV1.1 (*M. tuberculosis*) 11896261 (*M. avium*) 6224275 (*M. marinum*). **C** Dose-response curves of *M. abscessus* ATCC 19977 (black circles), D436H mutant (red squares), *M. abscessus* empty vector (white diamonds), *M. abscessus* pMVhsp60_leuS (pink triangles) and *M. abscessus* pMV306hsp60_leuSD436H (crosses). Data shown is mean ± SD from one experiment representative of two. **D** *leuS* is an essential gene in *M. abscessus* using CRISPRi gene knockdown. *M. abscessus* carrying integrated empty CRISPRi vector (EV, circles), CRISPRi:*mmpL4b* as a non-essential gene control (squares), or CRISPRi:*leuS* (triangles) were grown with/without 0.01 µg/mL ATc. Data is mean ± SD from biological triplicates. **E** MIC$_{99}$ to EPT and RFB in CRISPRi knockdown of *leuS* in *M. abscessus*. **, p = 0.005; ns, no statistical significance by one-way ANOVA with Dunnett's multiple comparisons test. Data shown is mean ± SD from three biological replicates.

when induced with anhydrotetracycline (ATc). We used CRISPRi to knockdown *leuS* which resulted in hyper susceptibility to EPT. Since *leuS* is an essential gene in *M. abscessus* (Fig 3D), we used 10-fold and 20-fold less ATc than required for complete growth suppression when

targeting an essential gene. The EPT $MIC_{99}$ of the strain induced with 0.1 μg/mL ATc decreased 10-fold compared to uninduced conditions, while induction did not affect the rifabutin (RFB) $MIC_{99}$ (Fig 3E). These results provide further evidence that EPT targets LeuRS in *M. abscessus*.

### EPT binds the editing active site of LeuRS

Although EPT is active against *M. abscessus* and *M. tuberculosis*, the $MIC_{90}$ for *M. abscessus* whole-cell activity is 7-fold lower ($MIC_{90Mabs}$ 0.063 μg/mL vs $MIC_{90Mtb}$ 0.46 μg/mL). We hypothesized that EPT had a higher affinity for the *M. abscessus* LeuRS editing active site. We therefore performed binding studies between EPT and the editing domains of LeuRS from *M. abscessus* and *M. tuberculosis* using isothermal titration calorimetry (ITC) (S4 Fig). *M. abscessus* LeuRS and *M. tuberculosis* LeuRS contain a single binding site for EPT, and we obtained similar equilibrium dissociation constants and Gibbs free energies (Table 2). The EPT—*M. abscessus* LeuRS binding has a higher enthalpic contribution (1.1 to 3.2 kcal $mol^{-1}$), while the EPT—*M. abscessus* LeuRS binding has a lower entropic contribution (5.5 to 3.3 kcal $mol^{-1}$).

To gain insight into the interactions between EPT and LeuRS, we solved the crystal structure of the *M. abscessus* editing domain ($LeuRS_{303-498}$). We obtained the crystal structures of the LeuRS editing domain unliganded 2.1 Å resolution (PDB 7N11) and in complex with the adenosine monophosphate (AMP) adduct with EPT at 1.7 Å resolution (PDB 7N12, S7 Table). AMP acts as a surrogate for the 3' end of the tRNA acceptor stem. In concordance with the binding mode of action of benzoxaboroles [19], we detected strong electron density in the active site corresponding to the EPT-AMP adduct formed through covalent interactions between the boron atom of EPT and the 2' and 3' hydroxyl groups on the ribose ring of AMP (Figs 4A and S5A). Comparing unliganded and liganded residues in *M. abscessus* editing domain structures shows a sizable shift of residues 416–422, located around the adenosine pocket, upon drug binding, including ordering of Y421 to interact with the EPT-AMP adduct phosphate, and a decrease of B factors in the neighbouring residues (S5B Fig). This increase in order could explain the relatively lower entropic contribution to binding observed in ITC, although an unliganded *M. tuberculosis* structure is not available for comparison.

Once bound with benzoxaborole-AMP adduct, the active sites of *M. tuberculosis* (PDB: 5AGR) [19] and *M. abscessus* LeuRS, and mode of adduct binding are very similar (Figs 4A and S5C2). Both proteins make critical contacts with the primary amine side chain through D433 (D447 in *M. tuberculosis*) and the universally conserved D436 (D450 in *M. tuberculosis*) (Fig 4A). In addition, R435 (R449 in *M. tuberculosis*) hydrogen bonds with the ethoxy oxygen of EPT and packed with the ribose of AMP (Fig 4A). The packing interaction on the ribose of AMP is underscored from *E. coli* mutants with R435H variants that lead to resistance early in the phase 2 clinical trial for complicated urinary tract infections. EPT has a hydroxypropyl ether moiety, rather than the ethyl ether in the benzoxaborole bound to *M. tuberculosis* LeuRS, but the extension shows very weak electron density and likely does not contribute to binding. The only notable difference is a ~1 Å shift in phosphate group of the EPT-AMP adduct (Fig

**Table 2. Thermodynamic analysis of EPT binding with LeuRS.**

| Bacteria | ΔG (kcal $mol^{-1}$) | ΔH (kcal $mol^{-1}$) | -TΔS (kcal $mol^{-1}$) | Kd (μM) |
|---|---|---|---|---|
| *M. abscessus* | -6.49 ± 0.07 | -3.2 ± 0.4 | -3.3 ± 0.3 | 16 ± 4 |
| *M. tuberculosis* | -6.9 ± 0.2 | -1.1 ± 0.3 | -5.5 ± 0.5 | 10 ± 4 |

ΔG, change in Gibbs free energy; ΔH, change in enthalpy; ΔS, change in entropy; T, temperature (303K)

Kd, dissociation constant

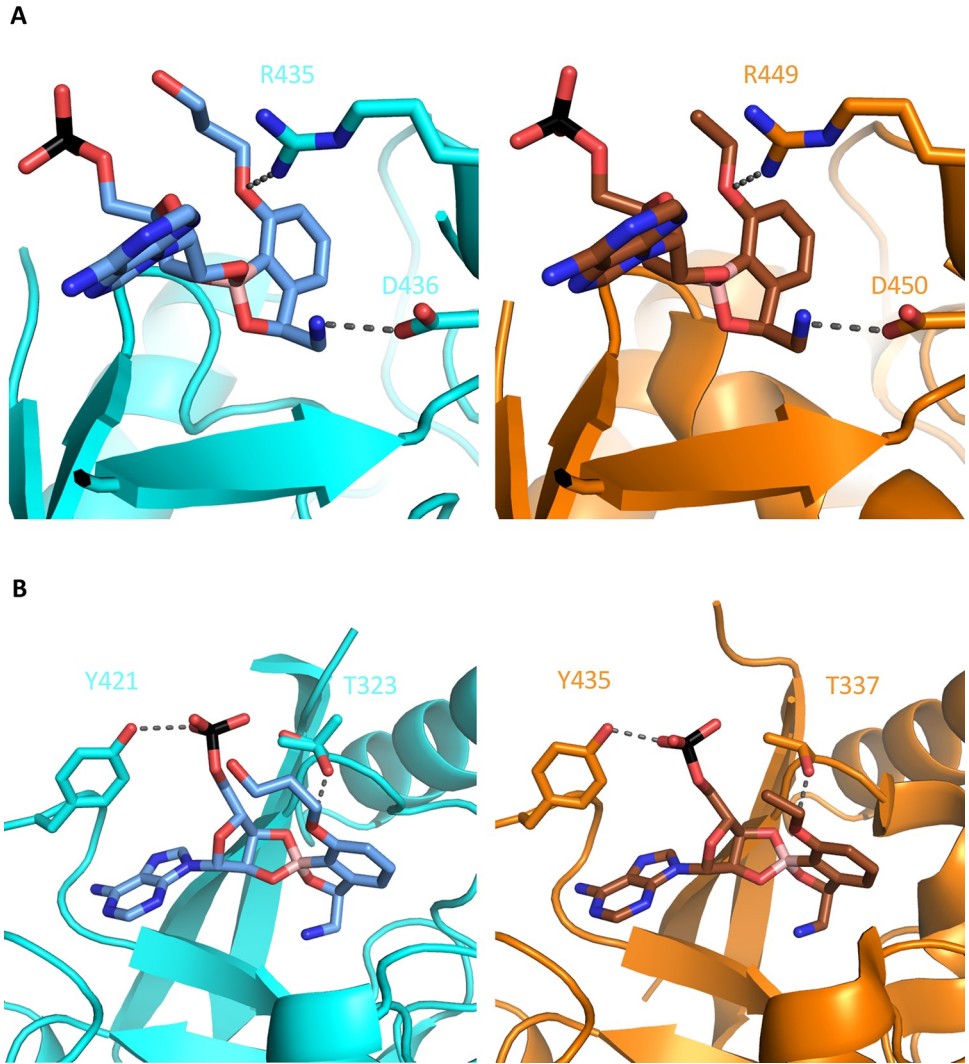

**Fig 4. Similar binding of benzoxaboroles to the editing domain of LeuRS.** EPT-AMP (blue) bound in the active site of *M. abscessus* LeuRS (cyan) and a similar benzoxaborole-AMP (brown) bound in active site of *M. tuberculosis* LeuRS (PDB 5AGR) (orange). **A** Critical arginine and aspartic acid residues make similar interactions with benzoxaboroles. **B** Phosphate from AMP pivots towards threonine residue in *M. abscessus* LeuRS. Conformer 1 of T323 points OH towards EPT, while conformer 2 orients OH towards the phosphate.

4B). In both complexes, the phosphate is pinned between Y421 (Y435) and T323 (T337). The electron density indicates that there are multiple rotameric conformations of T323, meaning its hydroxyl could hydrogen bond with the phosphate or with the 3' O and ether oxygen of the adduct (Fig 4B). *M. tuberculosis* LeuRS T337 is modelled in the latter position, but inspection of deposited maps also indicates multiple rotameric conformations. Variants at T322, T323 and T327 were identified in *P. aeruginosa* and *E. coli* mutants resistant to benzoxaboroles, which supports the importance of the threonine rich region for editing activity [32]. From the structural data, we propose that EPT binds LeuRS from *M. abscessus* and *M. tuberculosis* in a similar manner with similar affinity, providing a shared pathway for future structure-activity relationship analysis.

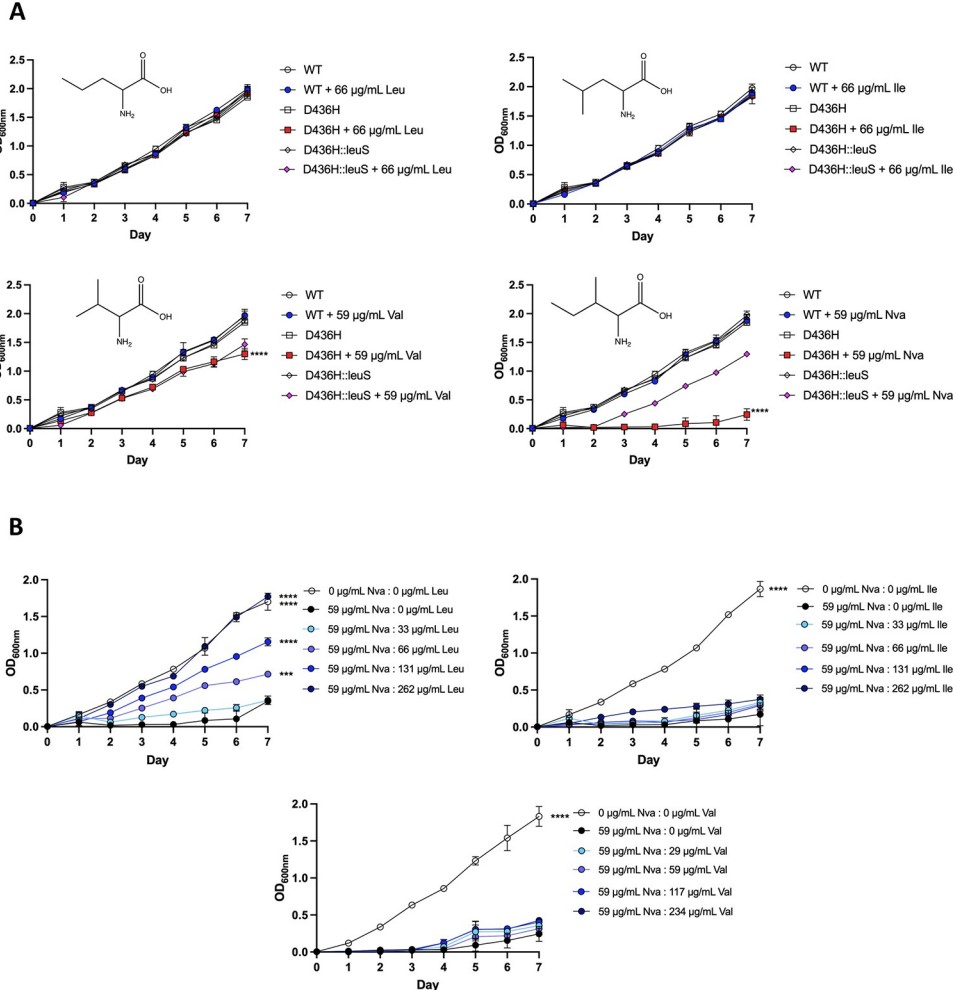

**Fig 5. Norvaline is toxic to *M. abscessus* D436H editing deficient mutants. A** ATCC 19977, D436H mutant, and D436H:leuS^D^ complement strains were grown in Sauton media with 59 μg/mL norvaline (top left), 66 μg/mL leucine (top right), 59 μg/mL valine (lower left), or 66 μg/mL isoleucine (lower right). Data shown is from three biological replicates, one-way ANOVA with Tukey's multiple comparisons test. ****, p < 0.0001. **B** D436H mutant grown in Sauton media with 59 μg/mL norvaline and a range of leucine (left), isoleucine (middle), or valine (right). D436H mutant grown in 0 μg/mL norvaline and 0 μg/mL BCAA represents growth control. D436H mutant grown in 59 μg/mL norvaline and 0 μg/mL BCAA represents inhibited growth control. BCAAs added to cultures from 33 μg/mL to 262 μg/mL (Leu/Ile) or 29 μg/mL to 234 μg/mL (Val). Data shown is mean ± SD from three biological replicates. Groups compared to inhibited growth control with one-way ANOVA with Dunnett's multiple comparisons test. ***, p = 0.0009; ****, p ≤ 0.0001.

## Norvaline is toxic to editing-deficient EPT mutants

Since resistance to EPT was shown in a phase 2 clinical trial of complicated urinary tract infections from *E. coli*, we sought a way to minimize the emergence of resistance in *M. abscessus* pulmonary infections [30]. We hypothesized that norvaline, a non-proteinogenic amino acid absent from extant proteins [33], would be toxic to *M. abscessus* editing-deficient mutants that acquired EPT resistance. When challenged with 59 μg/mL norvaline, the D436H editing-deficient mutant was significantly reduced in growth, while the wild-type strain was not inhibited (Fig 5A). The mutant strain complemented with wild-type *leuS* (D436H::leuS^D^) displayed a two-day lag before significant growth. As controls, leucine and isoleucine were not toxic to the

D436H mutant. There was however a minor reduction in mutant growth in 59 μg/mL valine, but no growth difference between the D436H and D436H::leuS$^D$ strains. Thus, we cannot conclude that valine exerts some toxicity to editing-deficient strains. Next, we asked if norvaline toxicity could be rescued with branched-chain amino acids (BCAAs). The D436H mutant was grown in 59 μg/mL norvaline with varying concentrations of BCAAs. Leucine but not isoleucine or valine rescued growth (Fig 5B). These observations corroborate the results that LeuRSs contain natural selectivity with their aminoacylation site for leucine and can discriminate against isoleucine and valine, but to a lesser extent against norvaline as non-cognate amino acids [34].

## Norvalination of the proteome induces the heat shock response

To provide a mechanism of norvaline toxicity, we analyzed the proteomes of wild-type, D436H mutant, and D436H:leuS$^D$ grown in norvaline or valine as control. We analyzed ~2500 proteins from whole-cell lysates after 12 hours or 3 days of incubation in 59 μg/mL norvaline or valine using reverse-phase HPLC/MS. Proteins filtered for leucine residues 14 Da lighter (14 Da corresponds to missing methylene group on norvaline) were enumerated using spectral counting. The wild-type maintained preferential incorporation of leucine into proteins, while the D436H mutant had a median norvaline misincorporation in 6% of the proteome after 12 hours. However, the amount of norvaline misincorporation between the D436H mutant and the D436H::leuS$^D$ strain was not significantly different (Fig 6A). Given that there was a two-day delay when the complemented strain was grown in norvaline (Fig 5A), we hypothesized that the effects of gene complementation would be seen at a later time point when grown in 59 μg/mL norvaline. The experiments were repeated, and the strains were incubated for 3 days. Again, the median norvalination increased in the D436H mutant relative to wild-type, but the level of misincorporated norvaline increased to 20% of the proteome. At this time point, the complemented strain partially rescued the level of norvalination after 3 days (Fig 6B). In all conditions tested, we measured background misaminoacylation of valine in the proteome (Fig 6A and 6B).

Next, we asked how the cell responded to stress caused by norvaline misincorporation in the proteome. Using total spectral counting, we determined the fold change in protein abundance from D436H mutant relative to wild-type. There was a distinct increase in proteins when challenged with norvaline that was absent from the valine control (Fig 6C-F). We used the protein-protein interaction (PPI) mapping tool STRING to identify upregulated proteins with common functions. When 130 of the most highly abundant proteins after norvaline stress were mapped for PPIs (Fig 7A), proteins belonging to the Clp protease family (ClpP1, ClpP2, ClpX) and GroEL chaperonin family (GroEL, GroL2, GroS) were identified (False discovery rate $7.2 \times 10^{-4}$ at 12 h, $2.1 \times 10^{-5}$ at 3 d) (Fig 7C). As a control, 130 randomly selected proteins from the norvaline dataset were not enriched for chaperonins and proteases when analyzed for PPIs (Fig 7B). We also observed a temporal change in chaperonin and protease levels where early stress from norvaline at 12 hours upregulated chaperonins followed by protease upregulation at 3 days (Fig 7D). This data suggests that norvaline misincorporation into the proteome results in toxicity from misfolded proteins [14,33,35–40].

## Norvaline reduces resistance to EPT *in vitro* and potentiates EPT activity *in vivo*

Knowing that norvaline targets LeuRS editing domain, we asked if norvaline could prevent escape mutants to EPT *in vitro*. We observed a 23-fold reduction in escape mutant frequency when *M. abscessus* ATCC 19977 was plated on 7H10 plates containing 10X MIC$_{90}$ EPT and

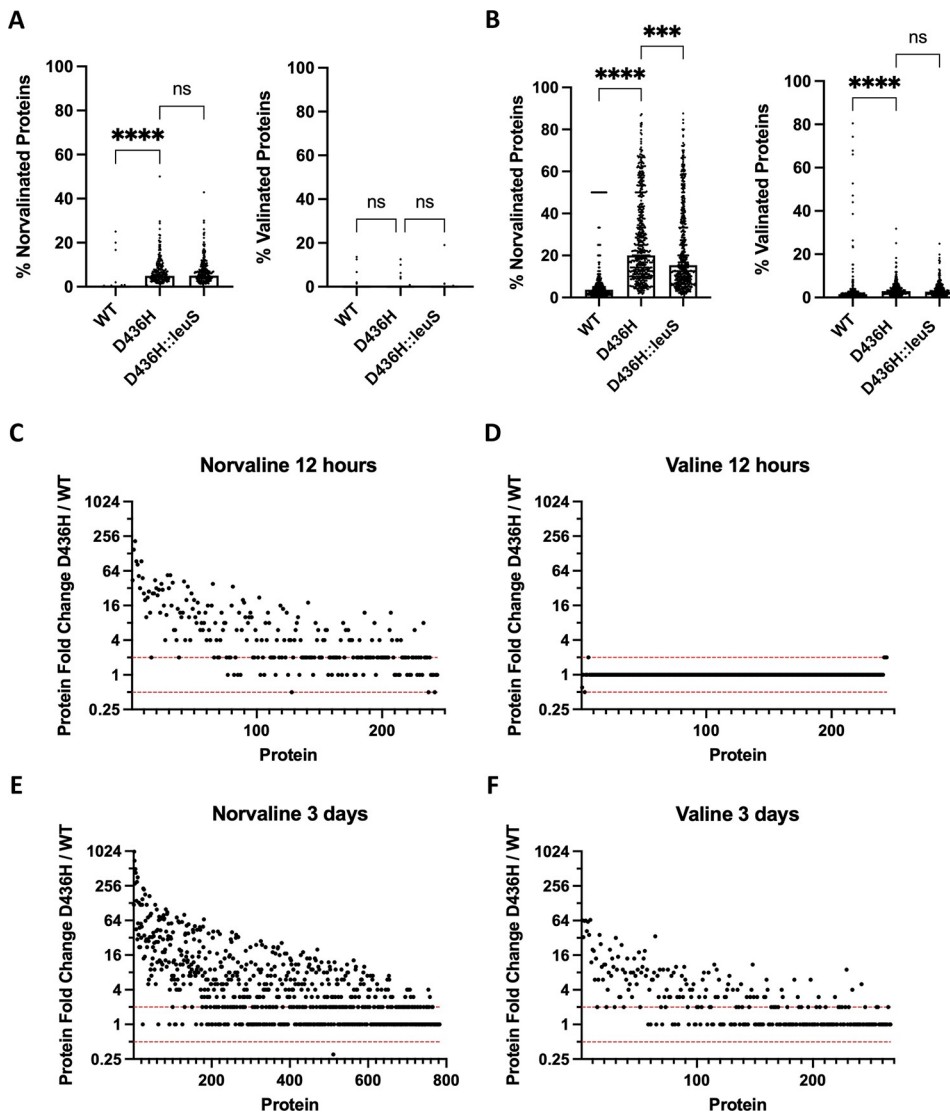

**Fig 6. Norvaline misaminoacylation in editing deficient mutants alters the proteomic landscape. A-B** *M. abscessus* ATCC 19977 or D436H mutant was grown in 59 µg/mL valine or 59 µg/mL norvaline for 12 hours or 3 days. Valine was used as a specificity control. Total cell lysate was collected. (Nor)valinated protein medians from the proteomes were compared using Kruskal-Wallis with Dunn's multiple comparisons test. ***, p = 0.0005; ****, p < 0.0001; ns, no statistical significance. **C-F** Protein fold abundance shown as D436H mutant relative to *M. abscessus* ATCC 19977. Dashed red lines indicate 0.5x and 2x fold abundance thresholds for biologically relevant changes.

590 µg/mL norvaline (1.89 x 10$^{-10}$) over EPT alone (4.28 x 10$^{-9}$) (Fig 8A); as a control, RFB did not benefit from norvaline supplementation. To test whether this effect would be observed in other mycobacteria, we repeated with *M. tuberculosis*, again observing a suppression of EPT mutants with norvaline exposure (Fig 8A). Next, the MIC$_{90}$ of EPT against *M. abscessus* ATCC 19977 was measured with differing doses of norvaline (Fig 8B). Norvaline had no effect on the MIC$_{90}$ even up to 590 µg/mL. These results suggest norvaline does not act as a traditional adjuvant to EPT.

We evaluated EPT and norvaline combination therapy for *in vivo* activity in a NOD.SCID mouse model of *M. abscessus* infection [41]. Mice were infected intranasally with a high

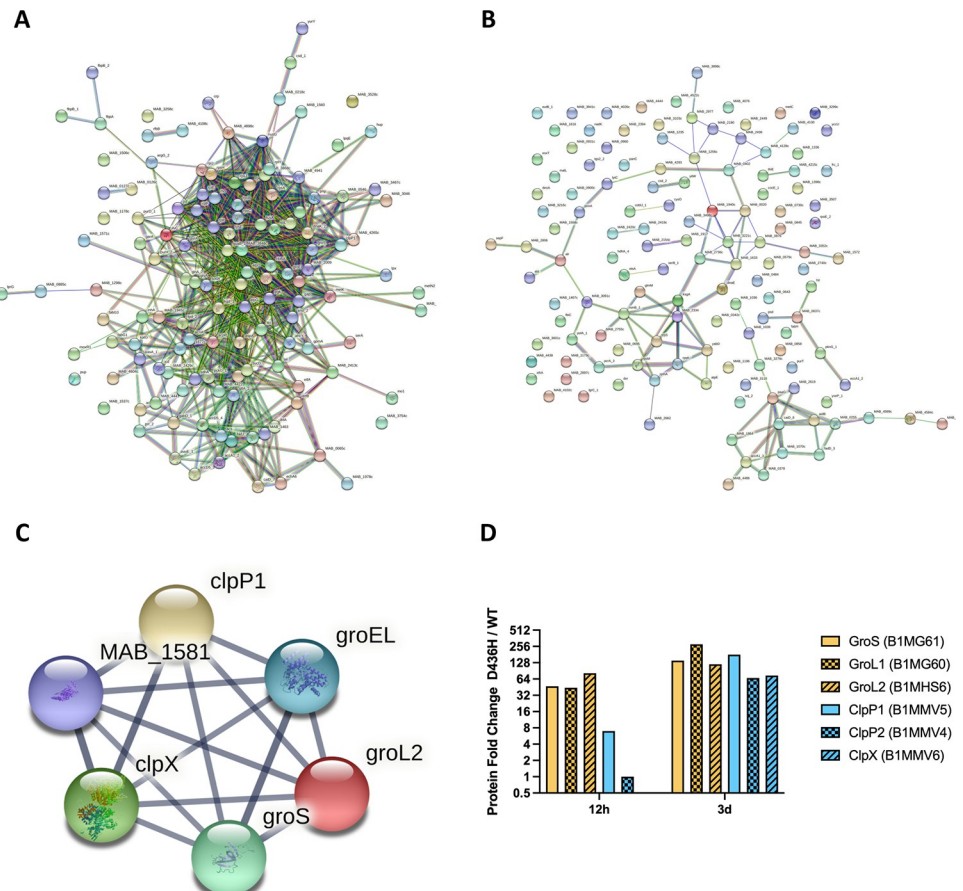

**Fig 7. Norvaline stress induces the heat shock response in editing deficient mutants. A** Protein-protein interaction network from the 130 most abundant proteins in D436H mutant when challenged with 59 μg/mL norvaline for 12 hours. 942 PPIs identified with PPI enrichment p-value $< 1.0\text{x}10^{-16}$. **B** PPI network from 130 randomly selected proteins in D436H mutant when challenged with 59 μg/mL norvaline for 12 hours. 102 PPIs identified with PPI enrichment p-value of 0.0397. **C** Clp protease/chaperonin family local network cluster identified in **a** (false discovery rate: 0.00072). **D** Temporal signature of the heat shock response. Protein expression levels of D436H norvaline relative to WT norvaline were measured by spectral counting.

inoculum of *M. abscessus* (~$10^6$ CFU). Treatment was started 1-day post-infection with once-daily oral vehicle (carboxymethylcellulose), RFB as positive control (10 mg/kg), EPT (10 mg/kg), EPT + norvaline (10 mg/kg + 3.3 mg/kg), or norvaline (3.3 mg/kg). Norvaline was dosed at 3.3 mg/kg (5 mM) which represents ~10X MIC$_{90}$ *in vitro* against the editing deficient D436H mutant. Previously, norvaline had been shown to be an effective *in vivo* neuroprotective agent as an arginase inhibitor at 2 mM in a triple-transgenic mouse model of Alzheimer's disease [42]. Compared to vehicle-treated mice, we measured a 1 log$_{10}$ decrease in bacterial burden in the lungs from RFB or EPT treatment. Furthermore, the addition of norvaline to the EPT group was significantly more active than EPT alone with an additional ~1 log$_{10}$ decrease in bacterial burden in the lungs while norvaline alone had no significant effect (Fig 8C).

## Discussion

Our data and two recent publications [43,44] confirmed the activity of benzoxaboroles against mycobacteria using zebrafish and murine models of infection and we showed that benzoxaboroles

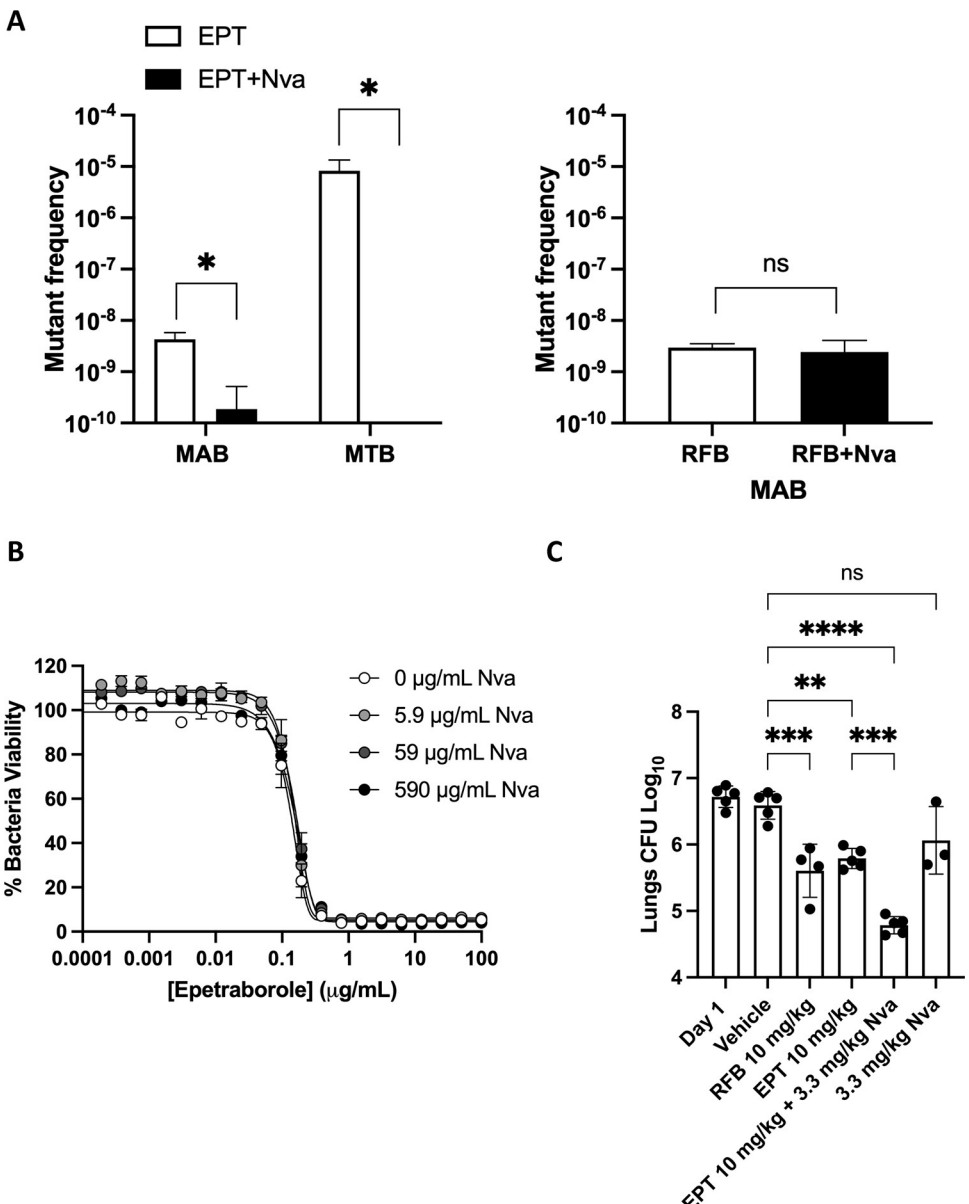

**Fig 8. Norvaline improves epetraborole efficacy *in vitro* and *in vivo*. A** Mutant frequency to 10X MIC$_{90}$ EPT (0.63 μg/mL) or 10X MIC$_{90}$ EPT + 10X MIC$_{90}$ norvaline (590 μg/mL) in *M. abscessus* ATCC 19977 and *M. tuberculosis* H37Rv. Data shown is mean ± SD from biological triplicates. Means compared using Poisson distribution. *, p < 0.0001; ns, no statistical significance. **B** *M. abscessus* susceptibility to EPT with norvaline as an adjuvant. Data shown is mean ± SD from one experiment representative of three. **C** EPT *in vivo* activity in SCID mouse model of *M. abscessus* lung infection with norvaline supplementation. Rifabutin (RFB) acts as positive control. Data shown is mean ± SD of three to five mice per treatment group. Means compared using one-way ANOVA with Tukey's multiple comparisons test. **, p < 0.007; ***, p < 0.0005; ****; p < 0.0001; ns, no statistical significance.

can be potentiated when combined with norvaline. Specifically, we demonstrated that the combination of EPT with norvaline reduces the emergence of *M. abscessus* and *M. tuberculosis* mutants and results in increased activity *in vivo* compared to EPT. Whether norvaline can be a therapeutic adjunct for other benzoxaboroles remains to be established, but is supported by observations with tavaborole and norvaline in *E. coli* [45]. The potential for combination therapy may be of value in

non-mycobacterial applications, such as the treatment of urinary tract infections, where EPT did not progress beyond phase 2 studies owing to the rapid emergence of resistance [30].

While we documented that EPT is active against both *M. abscessus* and *M. tuberculosis in vitro*, unexpectedly, we observed that EPT is more active against *M. abscessus*, which conflicts with the standard experience in antimycobacterial drug discovery [6,46–51]. Our cocrystal structure and ITC data point to similarities and differences in the interaction of EPT with the respective LeuRS proteins. Most notably, while the Kd values were similar, the enthalpy was greater for the EPT interaction with LeuRS of *M. abscessus*. Knowing that compounds with more enthalpically favourable binding are prioritized in hit-to-lead optimization [52], it may be possible that the discordance in enthalpy in part explains the difference in biological activity of EPT against the two organisms ($MIC_{90Mabs}$ 0.063 μg/mL vs $MIC_{90Mtb}$ 0.46 μg/mL). Other possibilities explaining the difference in whole-cell activity may be linked to differential expression level of *leuS* in both species, the lack of efflux pumps in *M. abscessus*, or the inability of *M. abscessus* to survive conditions that mimic leucine starvation, although these remain to be addressed. At a minimum, the current data can be used for future structure-activity relationship studies aimed at uncovering analogues with improved potency against LeuRS of *M. abscessus*, *M. tuberculosis* and other mycobacteria.

Our *in vivo* data shows the value of EPT alone in rescuing zebrafish embryos from lethal *M. abscessus* infection and that EPT + norvaline can control experimental pulmonary *M. abscessus* infection to a greater degree than the control antibiotic, RFB. Rather than acting as a traditional adjuvant like *β*-lactamase inhibitors for *β*-lactams [53], we propose that norvaline potentiates EPT *in vitro* activity by maintaining a pharmacological pressure on *M. abscessus* to remain editing proficient and limit the toxicity induced by the unfolded protein response and thus, limiting escape mutants. However, given that our *in vivo* model of pulmonary *M. abscessus* infection in NOD.SCID mice does not exceed a bacterial burden of $\sim 10^7$ in the lungs and that the rate of resistance against EPT is $\sim 10^{-9}$, other mechanisms might better explain our findings. Since the NOD.SCID model is immunocompromised for NK cells and adaptive immunity, macrophages—the most notable front-line defence for mycobacterial pulmonary infections—would be the main immune cell responsible for controlling the *M. abscessus* infection. One of the primary mechanisms of macrophages for controlling infections is the production of reactive nitrogen species like nitric oxide (NO) from inducible nitric oxide synthase (iNOS). Because NO is produced from arginine by iNOS, macrophages must regulate arginine metabolism between iNOS and the urea cycle, which naturally metabolizes arginine through the enzyme arginase. Interestingly, norvaline was shown to be an inhibitor of arginase and this resulted in increased production of NO in activated macrophages [54,55]. This increased production of NO from arginase inhibition might explain the lower bacterial burden observed in the NOD.SCID mouse group that received EPT + norvaline treatment.

Given that response rates to *M. abscessus* treatment are poor, resulting in chronic and potentially untreatable pulmonary and dermatologic infections, we submit that benzoxaboroles with the addition of a leucine mimic hold promise for the treatment of *M. abscessus* infections, and potentially other mycobacterial infections, such as tuberculosis.

## Materials and methods

### Ethics statement

All protocols involving mice followed the guidelines of the Canadian Council on Animal Care (CCAC) and were approved by the ethics committees of the RI-MUHC (project identifier [ID] 2015–7656).

## Culture conditions

Mycobacteria strains were grown in rolling liquid culture at 37˚C in Middlebrook 7H9 (Difco) supplemented with 10% albumin dextrose catalase (ADC), 0.2% glycerol, and 0.05% Tween 80 (7H9 complete) or on 7H10 agar plates supplemented with 10% oleic acid ADC and 0.5% glycerol at 37˚C. When mentioned, the carbon source was modified from glycerol to acetate or the detergent Tween 80 was removed from the media. *Bacillus cereus* (clinical isolate), *Corynebacterium glutamicum* (ATCC 13032), *Escherichia coli*, and *Pseudomonas aeruginosa* PAO1 were grown in LB broth. HepG2 cells (ATCC HB-8065) were grown in Dulbecco's modified Eagle's medium (DMEM) with phenol red (from Gibco) supplemented with 10% fetal bovine serum at 37˚C with 5% $CO_2$.

## Compound libraries

The GSK library contains 176 small molecules with antimycobacterial activity, specifically against *Mycobacterium tuberculosis* [56]. Medicines for Malaria Venture created two compound sets: the Pathogen Box and the Pandemic Response Box. The Pathogen Box is composed of 400 molecules active against various disease sets (https://www.mmv.org/mmv-open/pathogen-box/about-pathogen-box). The Pandemic Response Box is composed of 400 diverse drug-like molecules broadly categorized as antibacterials (201 molecules), antivirals (153 molecules), and antifungals (46 molecules) (https://www.mmv.org/mmv-open/pandemic-response-box/about-pandemic-response-box).

## Library screening

MMV Pathogen Box, MMV Pandemic Response Box, and GSK TB-active libraries were screened against *M. abscessus* ATCC 19977 strain which constitutively expressed the *luxCDABE* operon (*M. abscessus* lux) at 10 μM in duplicate in 96-well flat-bottom plates in 7H9 complete media. The culture was grown to log phase ($OD_{600}$ 0.4–0.8) and diluted to $OD_{600}$ 0.005 (5x10$^6$ CFU/mL). 90 μL of culture was mixed with 10μL of compound. Plates were sealed with parafilm and incubated at 37˚C for 48 hours. Plates had a column of 0.1% DMSO as negative controls (drug-free conditions) and 64 μg/mL of AMK as positive controls. Luminescence was measured with an Infinite F200 Tecan plate reader. % Luminescence relative to DMSO control was plotted using GraphPad Prism version 9. Compounds that decreased luminescence ≤ 10% were classified as primary hits. Primary hits were screened against *M. abscessus* ATCC 19977 at 10 μM in triplicate using REMA (see "Determination of MIC" below). Plates were setup as performed for the primary screen. Fluorescence was measured with an Infinite F200 Tecan plate reader. % bacteria viability relative to DMSO control was plotted using GraphPad Prism version 9. Compounds that decreased bacteria viability ≤ 10% were classified as confirmed hits. Confirmed hits were followed up with dose-response curves to determine the minimum inhibitory concentration (MIC).

## Determination of minimum inhibitory concentrations (MIC)

MIC values were determined using the resazurin microtiter assay (REMA). Cultures were grown to log phase ($OD_{600}$ of 0.4–0.8) and diluted to $OD_{600}$ of 0.005. Drugs were prepared in two-fold serial dilutions in 96-well plates with 90 μL of bacteria per well to a final volume of 100 μL. Plates were incubated at 37˚C until drug-free wells were turbid (2 days for *M. abscessus*). Ten μL of resazurin (0.025% wt/vol) was added to each well. Once the drug-free wells turned pink (one doubling time), the fluorescence (ex/em 560nm/590nm) was measured using an Infinite F200 Tecan plate reader. Fluorescence intensities were converted to % viable cells relative to drug-free conditions and fit to the Gompertz equation using GraphPad Prism

version 9. MIC values at 90% growth inhibition were determined from the nonlinear regression Gompertz equation.

## In vitro cytotoxicity in HepG2 cells

Drugs were prepared in two-fold serial dilutions in 96-well plates with 45 μL of Human HepG2 cells (2,000 cells/well) to a final volume of 50 μL. Plates were incubated for 3 days at 37˚C with 5% $CO_2$. Five μL of resazurin (0.025% wt/vol) was added to each well and incubated for 4 hours at 37˚C. Cell viability was determined from the fluorescence intensity as done in the previous REMA assay. 50% toxic dose concentrations ($TD_{50}$) values were obtained using a nonlinear regression fit equation ([inhibitor] vs response, variable slope) in GraphPad Prism version 9.

## Assessment of EPT efficacy in infected Zebrafish

Experiments in zebrafish were conducted according to the Comité d'Ethique pour l'Expérimentation Animale de la Région Languedoc Roussillon under reference 2020022815234677V3. Experiments were performed using the *golden* mutant and macrophage reporter Tg(*mpeg1:mCherry)* lines as previously described [57]. Embryos were obtained and maintained as described [27]. Embryo age is expressed as hours post fertilisation (hpf). Green fluorescent *M. abscessus* CIP104536[T] (R) expressing Wasabi were prepared and microinjected in the caudal vein (2–3 nL containing ≈100 bacteria/nL) in 30 hpf embryos previously dechorionated and anesthetized with tricaine [58]. The bacterial inoculum was checked *a posteriori* by injection of 2 nL in sterile PBS[T] and plating on 7H10[OADC]. Infected embryos were transferred into 6-well plates (12 embryos/well) and incubated at 28.5˚C to monitor kinetics of infection and embryo survival. Survival curves were determined by counting dead larvae daily for up to 11 days, with the experiment concluded when uninfected embryos started to die. EPT treatment of infected embryos and uninfected embryos was commenced at 24 hpi (hours post-infection) for 5 days. The drug-containing solution was renewed daily. Bacterial loads in live embryos were determined by anesthetising embryos in tricaine as previously described [59], mounting on 3% (w/v) methylcellulose solution and taking fluorescent images using a Zeiss Axio Zoom.V16 coupled with an Axiocam 503 mono (Zeiss). Fluorescence Pixel Count (FPC) measurements were determined using the 'Analyse particles' function in ImageJ [58]. All experiments were completed at least three times independently.

## Isolation of resistant mutants

Early log phase *M. abscessus* ($OD_{600}$ 0.1–0.4) was cultured and resuspended to an $OD_{600}$ of 10. Ten mL of culture was kept as a reference strain for future sequencing. One hundred μL of $OD_{600}$ 10 (approx. $1 \times 10^9$ CFU/100μL) was plated on solid media containing 10X, 20X, or 40X $MIC_{90}$ of the compound of interest. AMK was used as a control antibiotic. Plates were incubated for 5 days at 37˚C. Number of colonies were counted to obtain mutation frequencies. Resistant colonies were transferred into fresh media without antibiotics (to avoid the emergence of secondary mutations). The MIC (REMA method) was measured for the mutant against the compound of interest as well as a panel of reference compounds with different targets for negative controls.

## Sequencing resistant mutants

Genomic DNA (gDNA) was extracted from the 10 mL reference aliquot of *M. abscessus* harvested during mutant isolation and fresh cultures of mutant strains with confirmed resistance (REMA method) using the Qiagen QIAamp UCP Pathogen Mini kit with a modified mechanical lysis protocol. Pellet culture by centrifugation and resuspend in 590 μL of ATL buffer

containing the Dx reagent in a low-bind tube. Add 40 μL of proteinase K (20 mg/mL) and 20 μL of lysozyme (100 mg/mL) and incubate for 30 minutes at 56˚C under agitation. Transfer into a Pathogen Lysis Tube L and vortex twice using a FastPrep-24 instrument at 6.5 m/s for 45 s with a 5-minute incubation on ice in between. Transfer supernatant into fresh 2 mL low-bind tube. Follow manufacturer's instructions for sample prep with spin protocol. gDNA was quantified with Quant-iT PicoGreen dsDNA kit. *leuS* from ATCC 19977 and EPT mutants was sequenced with Sanger Sequencing with the primers listed in S5 Table.

### Cloning and over-expressing mutant *leuS* in *M. abscessus*

Wildtype and mutant *leuS* (D436H) were PCR amplified from gDNA using Phusion with primers 1 & 2 (see S5 Table), ligated into pMV306_hsp60 digested with EcoRV and HindIII restriction enzymes, and transformed into *E. coli* DH5α cells (Promega). Plasmids were extracted and sequenced with primers 3–12 (see S5 Table) using Sanger sequencing.

### Protein purification

The *M. abscessus* LeuRS editing domain (residues 303–499 of WP005112800.1) and the *M. tuberculosis* LeuRS editing domain (residues 311–512 of WP_003900794.1) were synthesized with an N-terminal polyhistidine tag and tobacco etch virus (TEV) protease recognition site, with codon optimization for *E. coli* and cloned into pET24a(+) by the company BioBasic to create pMabsED and pMtbED, respectively. *E. coli* BL21 (DE3) cells were transformed with pMabsED or pMtbED and grown overnight at 37˚C on LB-agar with 50 μg/mL kanamycin. Single colonies were transferred into 100 mL of LB media with 50 μg/mL kanamycin (LB-kan) and grown overnight at 37˚C. One liter of LB-kan was inoculated with 10 mL of the overnight culture and grown at 37˚C until an optical density at 600 nm of 0.6 was reached. Protein expression was then induced with addition of 0.5 mM IPTG and the culture further incubated for 18 h at 16˚C. The cells were harvested by centrifugation (4000 g for 20 min), resuspended in buffer A (50 mM Tris, 150 mM NaCl, 2 mM BME) plus 2 mM PMSF, lysed using sonication (55% amplitude, 30 cycles of 10s on 20s off) and the lysate clarified by ultracentrifugation (40,000 g for 30 min). The lysate was loaded onto a HiTrap FF (Cytiva) and eluted with buffer A plus 500 mM imidazole. The eluate was dialyzed in buffer A, digested with TEV protease overnight and the protease as well as non-cleaved protein separated from cleaved protein by application to the HiTrap FF column with flow-through collected. Protein was then subjected to size exclusion chromatography using a HiLoad Superdex Increase 75 colum (Cytiva), with fractions containing pure protein pooled.

### Isothermal titration calorimetry

ITC measurements were performed at 30˚C on a VP-ITC system (Malvern Panalytical Inc). Epetraborole (100 μM) in measurement buffer (50 mM Tris, 150 mM NaCl, 2 mM BME, 10 mM AMP) was titrated with protein solution (1 mM) in measurement buffer over 29 injections of 10 μl with 300 seconds equilibration time between injections. The heat evolved after each protein injection was calculated by integrating the calorimetric signal. The binding isotherms obtained were fitted to a single-site model using Origin 7 (Microcal Inc.). Experiments were performed in triplicates.

### Crystallography

Initial crystals of the *M. abscessus* LeuRS editing domain were obtained from sparse matrix screening in 96-well sitting drop format using a sample of 10 mg/mL protein in buffer A, and a

precipitant solution of 100 mM HEPES pH 7.0 and 2 M ammonium sulfate, at room temperature. Diffraction-quality crystals were grown by mixing 2 μL of protein solution (7.5 mg/mL) and 2 μL of reservoir solution (100 mM HEPES pH 7.0 and 2.5 M ammonium sulfate) in 24-well sitting-drop format. Crystals were cryoprotected in 100 mM HEPES pH 7.0 and 3.5 M ammonium sulfate, looped and flash-vitrified in liquid nitrogen. Diffracting co-crystals of the editing domain and epetraborole were obtained by streak seeding un-liganded crystals shards into drops in 24-well sitting-drop format with a reservoir solution of 100 mM HEPES, pH 7.5, 2% PEG400, 2.1 M ammonium sulfate, 10 mM AMP, 1 mM epetraborole and 15% glycerol at room temperature. These crystals were directly looped and flash-vitrified in liquid nitrogen for diffraction experiments.

## Structure determination

Diffraction data for the unliganded editing domain was collected at the Advanced Photon Source (24-ID-E) and data for the epetraborole–editing domain complex at the Canadian Light Source (CMCF-08B1-1). The data sets were indexed, processed, and scaled with HKL2000 [60] (for the unliganded editing domain) or DIALS [61] (for the complex). Initial phases for the *M. abscessus* LeuRS editing domain were obtained using molecular replacement with the program Phaser [62] using 5AGR [19] as a search model. Iterative rounds of refinement with Phenix [63] and manual modeling in Coot [64] yielded the final unliganded structure. The structure of the *M. abscessus* LeuRS editing domain with epetraborole bound in the active site crystallized in a different crystal form (S5 Table). Therefore, initial phases for the co-complex structure were obtained with molecular replacement using the program Phaser [62] utilizing the unliganded structure as a search model. The model for epetraborole was generated with eLBOW [65] and geometrical restraints were obtained with AceDRG [66]. Iterative rounds of refinement using Phenix [63] and manual modeling in Coot [64] yielded the final co-complex structure.

## Time-kill assays

Log phase ($OD_{600}$ of 0.4–0.8) *M. abscessus* was diluted to an $OD_{600}$ of 0.0001 ($10^5$ CFU/mL) and incubated with drugs of interest. One hundred μL aliquots were taken and serially diluted in 7H9 complete and plated on 7H10 agar. The starting inoculum was determined from time 0 before drugs were added. CFUs were determined after 4 days of incubation at 37˚C. Bactericidal activity is defined as a 3 $log_{10}$ decrease (99.9%) from the starting inoculum.

## Checkerboard assays

Drug combinations were assessed for synergy, indifference, or antagonism using the standard checkerboard format followed by REMA for MIC determination. Fractional inhibitory concentrations (FICs): FIC (X+Y) = (MIC of X in combination with Y)/(MIC of X alone). The fractional inhibitory concentration index (FICI) was calculated from $FIC_X + FIC_Y$. FICI values < 0.5 are defined as synergy, FICI values ≥ 4.0 are defined as antagonism, and FICI values in between are defined as indifferent. EPT was serially diluted two-fold from 4X MIC to 1/16 X MIC. Mycobacterial drugs were serially diluted two-fold from 8X MIC to 1/32X MIC.

## Inducible resistance assay

To test whether *M. abscessus* has inducible resistance to EPT, we performed a preexposure assay. Briefly, log phase *M. abscessus* was diluted to 0.05 and grown overnight with ¼ X $MIC_{50}$ of EPT (0.016 μg/mL) and CLR (0.016 μg/mL). After overnight growth, MICs were determined via REMA. Data is reported as the ratio of pre-exposed culture MIC to MIC of culture grown in drug-free conditions.

## CRISPRi conditional knockdown of *leuS*

The PLJR962 plasmid was restriction digested with BsmBI and gel purified. Synthetic oligo primers (S4 Table) for the sgRNA with the PAM sequence for *leuS* were annealed and ligated into digested PLJR962 vector and transformed into *E. coli* DH5α. Clones were sequenced using sequencing primer (S4 Table). 500 ng of CRISPRi *leuS* vector was electroporated into *M. abscessus* ATCC 19977. Colonies were picked from 7H10 plates containing 250 µg/mL kanamycin and confirmed by PCR. *M. abscessus* CRISPRi *leuS* was grown in liquid culture to early log phase (OD$_{600}$ 0.1–0.4) and diluted to 1x10$^4$ CFU/mL. One hundred µL of culture was plated on 7H10 containing either serial dilutions of EPT or RFB as control, and 0, 0.05 µg/mL, or 0.1 µg/mL ATc, the inducer of the sgRNA and catalytically inactive Cas9 (dCas9). The MIC$_{99}$ is the concentration of EPT or RFB that prevented growth relative to the drug free plate.

## Norvaline suppression of mutants

Early log phase *M. abscessus* or *M. tuberculosis* (OD$_{600}$ 0.1–0.4) was cultured and resuspended to an OD$_{600}$ of 10 (*M. abscessus*) or OD$_{600}$ of 1 (*M. tuberculosis*). One hundred µL of culture (approx. 1 X 10$^9$ CFU for *M. abscessus* and 1 x 10$^6$ CFU for *M. tuberculosis*) was plated on solid media containing 10X MIC$_{90}$ EPT, 10X MIC$_{90}$ EPT + 590 µg/mL norvaline, or 590 µg/mL norvaline. The experiment was repeated with 10X MIC$_{90}$ RFB as control. *M. abscessus* plates were incubated for 5 days at 37˚C, *M. tuberculosis* plates were incubated for 5 weeks at 37˚C. Number of colonies were enumerated to obtain mutation frequencies. Since, mutations are considered rare events, mutation frequency rates between 10X MIC$_{90}$ EPT and 10X MIC$_{90}$ EPT + 590 µg/mL norvaline were compared using the two Poisson rates.

The Poisson rate is defined as the number of events divided by the sample size: $\lambda = X/N$

The rates were compared using the small sample z-test:

$$z_{SR} = \frac{\sqrt{\lambda_2} - \sqrt{\lambda_1}}{\frac{1}{2}\sqrt{\frac{1}{N_1} + \frac{1}{N_2}}}$$

## LC-MS/MS measurement of norvaline in the proteome

Wildtype, mutant, and complement *M. abscessus* strains were grown for 12 h or 3 days in 59 µg/mL norvaline or 59 µg/mL valine. Proteins were extracted using an optimized protocol for mass spectrometry follow-up [67]. Cells were collected, washed with ice-cold PBS, and resuspended in 1mL lysis buffer (50 mM NH$_4$HCO$_3$ pH 7.4, 10 mM MgCl$_2$, 0.1% NaN$_3$, 1 mM EGTA, 1 x protease inhibitors (Roche), 7 M urea, and 2 M thiourea). Cells were lysed with zirconia beads and the cell lysate was collected and filtered through a 0.22 µm membrane. Proteins were precipitated overnight at 4˚C with TCA at 25% (v/v). The precipitate was washed with 1 mL cold acetone and 250 µL cold water. The final wash is only water. The pellet was resuspended in 200 µL resuspension buffer (50 mM NH$_4$HCO$_3$, 1 M urea). Protein extraction was quantified with the Bradford assay and the quality of proteins was examined on SDS-PAGE. Protein lysates were dissolved in SDS-PAGE reducing buffer and electrophoresed onto a single stacking gel band to remove lipids, detergents and salts. For each sample, a single gel band was reduced with DTT (Sigma), alkylated with iodoacetic acid (Sigma) and digested with LCMS grade trypsin (Promega). Extracted peptides were re-solubilized in 0.1% aqueous formic acid and loaded onto a Thermo Acclaim Pepmap (Thermo, 75 µM ID X 2 cm C18 3 µM beads) precolumn and then onto an Acclaim Pepmap Easyspray (Thermo, 75 µM X 15 cm with 2 µM C18 beads) analytical column separation using a Dionex Ultimate 3000 uHPLC at 250 nl/min with a gradient of 2–35% organic (0.1% formic acid in acetonitrile) over 2 hours. Peptides were analyzed using a Thermo Orbitrap Fusion mass spectrometer operating at

120,000 resolution for MS1 with HCD sequencing at top speed (15,000 resolution) for all peptides with a charge of 2+ or greater. The raw data were converted into *.mgf format (Mascot generic format) for searching using the Mascot 2.5.1 search engine (Matrix Science) against Mycobacterium abscessus protein sequences (Uniprot downloaded 2020.11.30). A modification for Xle->Val was used to detect incorporation of Val into WT sequences. The database search results were loaded onto Scaffold Q+ Scaffold_4.4.8 (Proteome Sciences) for statistical treatment and data visualization.

## Murine model of chronic *M. abscessus* infection using NOD.CB17-Prkdc$^{scid}$/NCrCrl mice

RFB (Sigma), EPT (Cayman Chemical), and norvaline (Sigma) were dissolved in 0.5% w/v sterile low viscosity carboxymethyl cellulose pH 7 (vehicle). Drugs were aliquoted and stored at -20°C for the duration of the 10-day treatment. 6–8 weeks-old female mice were ordered from Charles River Labs. Mice were intranasally infected with ~$10^6$ CFU (25 μL of 5x$10^8$ CFU/mL) of *M. abscessus* CIP 104536$^T$ (rough morphotype). Five mice were sacrificed 4 hours post-infection to determine initial lung inoculum and on day 1 to enumerate CFU prior to drug administration. Mice were treated daily by oral gavage with 100 μL of vehicle or drug. On day 11, mice were humanely euthanized, the lungs were collected and homogenized in 1mL of 7H9 complete. Lung homogenates were plated on 7H11 agar and plates were incubated at 37°C for 5 days. CFU data was log-transformed for one-way ANOVA with Tukey's multiple comparisons test using GraphPad Prism version 9.

## Supporting information

**S1 Fig. *M. abscessus* phenotypic screening with 176 TB-active compounds from GSK (A-C) and MMV (D-F). A** Luminescence relative to drug free conditions using *M. abscessus* lux. Data is shown in duplicate. **B-C** Secondary screening of primary hits using REMA on *M. abscessus* ATCC 19977. % bacteria viability relative to drug free conditions. Data is shown as mean ± SD from technical triplicates. Dashed red line indicates 10% viability threshold when screened at 10$\mu$M. Three hits were identified from primary screen however, all three did not pass the secondary screen. No active compounds were identified from this library. **D** MMV Pathogen box and **E** Pandemic response box; luminescence relative to drug free conditions using *M. abscessus* lux. Data is shown in duplicate. **F** Secondary screening of primary hits using REMA on *M. abscessus* ATCC 19977. % bacteria viability relative to drug free conditions. Data is shown in technical triplicate. Dashed red line indicates 10% viability threshold when screened at 10$\mu$M. **G** Correlation between hits from luminescence primary screen and bacterial viability secondary screen. Correlation calculated using non-parametric Spearman correlation. 20 compounds passed the primary screen and 9 passed the secondary screen. Only three compounds were still active after acquiring fresh batch and displayed dose-dependent activity (3/800). EPT in cyan. AMK in blue as positive control. Two negative hits from luminescence screen in magenta used as negative controls. Drug free in black used as media control. (TIFF)

**S2 Fig. Isobolograms for potential combination therapy with EPT.** Green area indicates synergy (FICI < 0.5); red area indicates antagonism (FICI $\geq$ 4.0); white area indicates indifferent. RIF and CLR, and AMK and CLR are used as synergy and antagonism controls, respectively. Data shown is from checkerboard assays done in technical duplicate. (TIFF)

**S3 Fig. Drug susceptibility assessment of resistant mutants. A-D** Dose-response curves of isolated EPT (10X, 20X, 40X MIC) mutants or AMK (40X MIC) mutant. AMK was used as control compound for mutant isolation. RIF and BDQ were used for cross-resistance verification. ATCC 19977 (black circles), D436H mutant-1 (yellow squares), D436H mutant-2 (pink diamonds), D436H mutant-3 (green triangles), D436H mutant-4 (purple inverted triangles), AMK 40X (crosses). Data is mean ± SD from two independent experiments.
(TIFF)

**S4 Fig. Thermodynamic analysis of EPT binding to LeuRS editing domain.** Heat of injection (upper panel) and single-site binding model of the integrated isotherm (lower panel). *Mabs* LeuRS (left) or *Mtb* LeuRS (right) editing domains bound to EPT.
(TIFF)

**S5 Fig. Co-crystal structure of LeuRS and benzoxaborole adducts. A** Difference maps for the EPT-AMP adduct in 7N12. Unbiased FO-FC difference maps ($2.8\sigma$), calculated with phases from a model that never included ligand. **B** Comparison of apo (green, PDB 7N11) and co-complex (cyan, PDB 7N12) structures of *M. abscessus* LeuRS bound to EPT-AMP. **C** Benzoxaborole inhibitors of LeuRS. (Left) EPT-AMP adduct bound to *M. abscessus* LeuRS. (Right) BNZ-AMP adduct bound to *M. tuberculosis* LeuRS.
(TIFF)

**S1 Table. Carbon dependent variability on EPT activity.** G, Glycerol; A, Acetate; Tw80, Tween-80; CaMH, Cation-adjusted Muller-Hinton.
(DOCX)

**S2 Table. Epetraborole *in vitro* activity against clinical isolates.** [a]Morphology as determined by smooth (S) or rough (R) colonies on 7H10 agar. Drugs used: EPT, Epetraborole; AMK, Amikacin; RIF, Rifampicin; BDQ, Bedaquiline; CFX, Cefoxitin; CLR, Clarithromycin.
(DOCX)

**S3 Table. Epetraborole *in vitro* activity spectrum.** [a]Acid fast (AF); gram positive (+); gram negative (-). EPT, Epetraborole; AMK, Amikacin; RIF, Rifampicin; ERV, BDQ, Bedaquiline; CFX, Cefoxitin; CLR, Clarithromycin. Species: *Mycobacterium abscessus*, *Mycobacterium avium hominissuis*, *Mycobacterium avium intracellulaire*, *Mycobacterium tuberculosis* H37Rv, *Mycobacterium tuberculosis* Erdman, *Bacillus cereus*, *Corynebacterium glutamicum*, *Escherichia coli*, *Pseudomonas aeruginosa*.
(DOCX)

**S4 Table. *In vitro* resistance frequency.** [a]N.D: not determined. [b]$MIC_{90}$ values on 7H10 agar used in this experiment were 4.0 μg/mL and 0.27 μg/mL for amikacin and epetraborole, respectively.
(DOCX)

**S5 Table. Primers.**
(DOCX)

**S6 Table. Whole genome sequencing variants identified in EPT-resistant *M. abscessus*.**
(DOCX)

**S7 Table. Data collection and refinement statistics.**
(DOCX)

## Acknowledgments

We thank Adam Hassan (Research Institute of the McGill University Health Centre) for technical assistance with mouse experiments. We thank Medicines for Malaria Venture for providing the MMV Open pathogen box and pandemic response boxes. *This research used resources of the Advanced Photon Source, a U.S. Department of Energy (DOE) Office of Science User Facility, operated for the DOE Office of Science by Argonne National Laboratory under Contract No. DE-AC02-06CH11357. Extraordinary facility operations were supported in part by the DOE Office of Science through the National Virtual Biotechnology Laboratory, a consortium of DOE national laboratories focused on the response to COVID-19, with funding provided by the Coronavirus CARES Act.* Part or all of the research described in this paper was performed using beamline CMCF-BM at the Canadian Light Source, a national research facility of the University of Saskatchewan, which is supported by the Canada Foundation for Innovation (CFI), the Natural Sciences and Engineering Research Council (NSERC), the National Research Council (NRC), the Canadian Institutes of Health Research (CIHR), the Government of Saskatchewan, and the University of Saskatchewan. The *luxCDABE* plasmid was kindly gifted by Jeffery S. Cox. We are grateful to the CRBM zebrafish facility (Montpellier), P. Richard and M. Plays for zebrafish husbandry.

## Author Contributions

**Conceptualization:** Jaryd R. Sullivan, Marcel A. Behr.

**Formal analysis:** Jaryd R. Sullivan, Elias Kalthoff, Claire Hamela, Lorne Taylor.

**Funding acquisition:** Marcel A. Behr.

**Investigation:** Jaryd R. Sullivan, Andréanne Lupien, Elias Kalthoff, Claire Hamela, Lorne Taylor, Kim A. Munro.

**Methodology:** Jaryd R. Sullivan, Elias Kalthoff, Claire Hamela.

**Project administration:** Marcel A. Behr.

**Resources:** T. Martin Schmeing, Laurent Kremer, Marcel A. Behr.

**Supervision:** Andréanne Lupien, T. Martin Schmeing, Laurent Kremer, Marcel A. Behr.

**Visualization:** Jaryd R. Sullivan, Claire Hamela, T. Martin Schmeing.

**Writing – original draft:** Jaryd R. Sullivan, Marcel A. Behr.

**Writing – review & editing:** Jaryd R. Sullivan, Andréanne Lupien, T. Martin Schmeing, Laurent Kremer, Marcel A. Behr.

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
