## [Decision Letter · Decision Letter 0]

12 Aug 2021

Dear Dr. Behr,

Thank you very much for submitting your manuscript "Efficacy of epetraborole against Mycobacterium abscessus is increased with norvaline" for consideration at PLOS Pathogens. As with all papers reviewed by the journal, your manuscript was reviewed by members of the editorial board and by several independent reviewers. The reviewers appreciated the attention to an important topic. Based on the reviews, we are likely to accept this manuscript for publication, providing that you modify the manuscript according to the review recommendations.

There are quite a substantial number of concerns, and for some, the authors may already have the data. Nevertheless, the concerns seem feasible enough to address as a minor revision.

Sincerely,

Helena Ingrid Boshoff

Associate Editor

PLOS Pathogens

JoAnne Flynn

Section Editor

PLOS Pathogens

Kasturi Haldar

Editor-in-Chief

PLOS Pathogens

orcid.org/0000-0001-5065-158X

Michael Malim

Editor-in-Chief

PLOS Pathogens

orcid.org/0000-0002-7699-2064

There are quite a substantial number of concerns, and for some, the authors may already have the data. Nevertheless, the concerns seem feasible enough to address as a minor revision.

Reviewer Comments (if any, and for reference):

Reviewer's Responses to Questions

**Part I - Summary**

Reviewer #1: The manuscript entitled "Efficacy of epetraborole against Mycobacterium abscessus is increased with norvaline" by Sullivan, describes the activity of epetraborate and norvaline against M. abscessus. Overall the paper is very interesting and generally well supported by experimental results. I think this paper could strongly benefit from enzymatic data. Structures look very good and support the results of the paper. There are several issues that should be addressed before the paper can be accepted.

1. The beginning of the abstract needs to be rewritten. The first few sentences are short and seem disjointed. It reads more like bullet points.

2. Line 103 – The authors used two different methods to measure viability in their whole cell active compound screen, namely luminescence and resazurin. I think it would be helpful to show the correlation between these two screens. I do not believe it changes validity of results, but it would be good to know how many compounds were active on luminescence, resazurin or both.

3. Line 114 – The author claims that EPT lost potency in cation-adjusted Muller Hinton. This seems like an important finding. I believe the authors would make a stronger statement if they could at least provide a at least some speculation on mode of action of EPT.

4. Line 144 – The author reference fig 1C. I found this figure to be challenging to navigate, due to the number of experiments being shown in each panel (MIC of EPT and CLR, with incubation on EPT or CLR, with and without Passage for 6 days). I found myself constantly having to go back to the legend when trying to compare datasets. I think it may be beneficial to split this figure in more panels to make it easier to understand

5. Fig 1C – on the 14 day figure, it seems like pre exposure to EPT, sensitized cells to both EPT and CLR. Especially when comparing the with and without 6 days of passage. The fold change is not large, but the small error bars for these data sets suggest it is statistically significant. Biological significance is however debatable.

6. Line 154 – The authors found no synergy between EPT and other antimycobacterial drugs, yet the text reads “indicating the potential for a future multidrug regime”. I understand they are mostly pointing out that EPT is at least not antagonistic, but I don’t know if these results demonstrate strong potential for combination therapy.

7. Line 161 – Wrong figure name. I believe this should be Fig 2A not 1A

8. Line 167 – The authors claim that there was only minimal benefit at 10ug/mL, but significantly improved at 40ug/mL. However, at day 11, the probability of survival is the same in both dosages. Significantly higher than untreated. Please explain.

9. Fig 2B – I believe this experiment should have been carried out longer. Although the 10ug trace started off closely matching the untreated, they separate in the later days. The authors repeatedly mention that abscessus is notoriously difficult to treat, with regimen taking 18 months, yet stopped recording data 11 days post infection. Zebrafish can live for 3 years. Furthermore at 11 days, survivability was still 40% in the untreated cohort, and was not showing sign of plateauing.

10. Line 185 – The authors conclude that the mechanism of resistance must be specific to EPT. However, they only test 3 other drugs. Although it is true that this seems EPT dependent, testing 3 other compounds, with unrelated mechanisms of action, is in my opinion not enough to say it is EPT specific.

11. Line 186 – The authors decided to use a focused approach to identify the mutation in leuS based on literature. However, with only 4 four mutants, I think doing whole genome sequence would have been feasible. It is entirely possible that each mutant had to accumulate more than the D436H mutation. Since the resistance frequency is higher than expected, this is a possibility.

12. Fig 3c – WT cells can become resistant by transforming with a mutant leuS, indicating the enzyme plays a role in the mechanism of resistance. However, the level of resistance is intermediate. I think this should be addressed in the paper to offer some hypotheses. Relative expression of WT leuS vs Mutant leuS in this strain could also explain the results. It is also possible the resistant mutant strain has other mutation contributing to the resistance.

13. Line 215 – The authors hypothesize that EPT is more potent against M. abscessus than M. tuberculosis because it has higher affinity to LeuRS active site in M. abscessus. Binding data does not strongly support that this is the case. However, in order to demonstrate stronger inhibition of the M. abscessus enzyme, I believe an enzymatic assay would be very important. The difference in whole cell potency between species could easily be due to difference in cell permeability, efflux pump or many other cellular processes that can affect potency of drugs. Aminoacyl-tRNA synthetase can be easily assayed by measuring released pyrophosphate or AMP.

14. PDB- 7N12 I could not access the structure however, the I/sigma is, I believe, too low in the PDB validation document at 0.2. I believe resolution should be cut.

15. Line 252 – Their final conclusion based on structure data seem to contradict their hypothesis that EPT has higher affinity for the M. abscessus enzyme. If this is the case, I believe authors should reflect back on the hypothesis of line 215.

16. Line 317 – Using Norvaline in order to reduce resistance is, in my opinion, very clever. However, I believe that in order to further demonstrate this as a possible avenue, the authors should either demonstrate themselves or provide evidence from literature that Norvaline is safe.

17. Line 337 – The authors are trying to make the argument that using Norvaline in combination with EPT would be valuable, because rapid emergence of resistance caused EPT not to progress in phase 2 clinical study (UTI). However, this contradicts their previous findings, where they claim that resistance rates were lower than expected in M. abscessus. The UTI trial was probably on a different organism, therefore emergence of resistance may not be a problem for treatment of m. abscessus.

18. Line345 – I do not think the enthalpy difference is strong enough to convincingly claim that EPT inhibits M. abscessus’ LeuRS more than Mtb’s. This needs to be assayed enzymatically.

Reviewer #2: I thought this was a good manuscript covering an important area of drug discovery for this unmet medical need. A large number of experiments have been performed and the article tells a very complete story. I would support acceptance of this manuscript with a few minor revisions highlighted below

Reviewer #3: This well written manuscript describes a series of experiments that identified EPT as a potent inhibitor of Mycobacterium abscessus, confirmed the expected targeting of LeuRS, demonstrated therapeutic proof of concept in a mouse model of M. abscessus infection and tested the novel hypothesis that adjunctive norvaline could potentiate EPT efficacy and suppress emergence of EPT resistance by compromising survival of EPT-resistant LeuRS editing domain mutants. The authors conclude that the approach of combining oxaborole LeuRS inhibitors with leucine mimics hold promise for treatment of M. abscessus infections and potentially tuberculosis and other NTM infections.

The work is novel, comprehensive and potentially impactful for several difficult-to-treat mycobacterial infections. In addition to EPT, the oxaborole LeuRS inhibitor GSK3036656 has progressed to a phase 2 trial for tuberculosis. A few major and minor issues are described in the comments below.

**Part II – Major Issues: Key Experiments Required for Acceptance**

Reviewer #1: (No Response)

Reviewer #2: None

Reviewer #3: The conclusion that the additive activity of norvaline and EPT observed in vivo is attributable to suppression of LeuRS mutants by the addition of norvaline is dubious. Given the extremely low frequency of spontaneous mutants observed in vitro, the limited growth of the bacteria in vivo and the limited bactericidal effect of EPT, these mutants are unlikely to have replaced the susceptible bacterial population in the lungs of mice after just 10 days of treatment. Thus, selective amplification of such mutants is unlikely to be the factor limiting the magnitude of the EPT effect in this model. This could be confirmed by testing the isolates obtained at the end of treatment for EPT resistance and LeuRS mutations. This comment is not intended to discount the observation of potentiation by norvaline but, rather, to challenge the proposed explanation. The proposed experiment could be avoided if the discussion is revised accordingly.

**Part III – Minor Issues: Editorial and Data Presentation Modifications**

Reviewer #1: (No Response)

Reviewer #2: Abstract, summary statement AND introduction should be reviewed to improve the flow of the text

The location of the resistant mutation (D436H) should be related to those seen in the clinical resistance in E.coli. In particular in relation to whether the same residues are seen and propose how the mutation causes resistance (in the crystallography section)

Is reference 40 the correct reference for 2mM in vivo treatment of norvaline, if it is then it needs to be made clear this its activity is not for mycobacterial infection models.

An explanation needs to proposed for why in vivo synergy is seen, I am not an expert in this area but in a study this short it seems unlikely to me that when there is no effect on MIC, synergy would be related to effects on escape resistance when only infecting with 106 bacteria and frequency of resistance is only 2 x-9

There are two publications this year relating to M abscessus and epetraborole. While this manuscript expands on these studies by looking at norvaline it should mention these earlier publications

Reviewer #3: The prospects for incorporating norvaline into clinical use for the adjunctive treatment of mycobacterial treatment could be discussed in more detail. For example, what is the relevance of the 5 mM concentration/dose tested in vitro and in vivo in light of what is known of the human exposures that may be safely attained and tolerated?

EPT concentrations are alternately described in uM or ug/ml in different sections of the manuscript; one unit should be adopted throughout, with the translation of uM to ug/ml encouraged if uM is used

Lines 157-70: in this paragraph, figs referenced should be figs 2a-d rather than figs 1a-d

Line 322: suggest revision to “~10X MIC90 in vitro against the D436H mutant” or something similar to clarify this MIC is against the mutant

Line 323: suggest revision to “effective in vivo as a neuroprotective agent” or something similar to clarify that this statement of efficacy applies to a different, non-infectious disease model

Line 386: please indicate which strain was engineered for luminescence

Lines 752-4: duplicate of ref 30

Lines 811-3: duplicate of ref 26

Fig 1 legend: for fig 1c, change “upper” and “lower” to “left” and “right”

PLOS authors have the option to publish the peer review history of their article (what does this mean?). If published, this will include your full peer review and any attached files.

Reviewer #1: No

Reviewer #2: No

Reviewer #3: No

Figure Files:

Data Requirements:

Reproducibility:

References:

---

## [Editor Report · Decision Letter 1]

23 Sep 2021

Dear Dr. Behr,

We are pleased to inform you that your manuscript 'Efficacy of epetraborole against Mycobacterium abscessus is increased with norvaline' has been provisionally accepted for publication in PLOS Pathogens.

Best regards,

Helena Ingrid Boshoff

Associate Editor

PLOS Pathogens

JoAnne Flynn

Section Editor

PLOS Pathogens

Kasturi Haldar

Editor-in-Chief

PLOS Pathogens

orcid.org/0000-0001-5065-158X

Michael Malim

Editor-in-Chief

PLOS Pathogens

orcid.org/0000-0002-7699-2064

The authors have addressed the reviewers' concerns. They were unable to express enzymatically active LeuRS. The measurement of inhibition of the enzyme would certainly have yielded interesting results but do not contribute the major findings of this work.
---

## [Editor Report · Acceptance letter]

6 Oct 2021

Dear Dr. Behr,

We are delighted to inform you that your manuscript, "Efficacy of epetraborole against </i>Mycobacterium abscessus</i> is increased with norvaline," has been formally accepted for publication in PLOS Pathogens.

Best regards,

Kasturi Haldar

Editor-in-Chief

PLOS Pathogens

orcid.org/0000-0001-5065-158X

Michael Malim

Editor-in-Chief

PLOS Pathogens

orcid.org/0000-0002-7699-2064